# Contributions of insula and superior temporal sulcus to interpersonal guilt and responsibility in social decisions

**Maria Gädeke[1], Tom Eric Willems[1,2], Omar Salah Ahmed[1,3,4], Bernd Weber[3,4], Rene Hurlemann[5], Johannes Schultz[3,4]\***

[1]Masters in Neuroscience Program, University of Bonn, Bonn, Germany; [2]Institute of Psychology, University of Bern, Bern, Switzerland; [3]Center for Economics and Neuroscience, University of Bonn, Bonn, Germany; [4]Institute of Experimental Epileptology and Cognition Research, Medical Faculty, University of Bonn, Bonn, Germany; [5]Department of Psychiatry, University of Oldenburg, Oldenburg, Germany

## eLife Assessment

This manuscript provides **valuable** novel insights into the role of interpersonal guilt in social decision-making by showing that responsibility for a partner's bad lottery outcomes influences happiness. Through the integration of neuroimaging and computational modelling methods, and by combining findings from two studies, the authors provide **solid** support for their claims. The findings will be of interest to researchers in the field of social neuroscience and decision making.

**\*For correspondence:**
johannes.schultz@ukbonn.de

**Competing interest:** The authors declare that no competing interests exist.

**Abstract** This study investigated the neural mechanisms involved in feelings of interpersonal guilt and responsibility evoked by social decisions in humans. In two studies (one during fMRI), participants repeatedly chose between safe and risky monetary outcomes in social contexts. Across conditions, each participant chose for both themselves and a partner (*Social* condition), or the partner chose for both themselves and the participant (*Partner* condition), or the participant chose just for themselves (*Solo* condition, control). If the risky option was chosen in the *Social* or *Partner* condition, participant and partner could each receive either the high or the low outcome of a lottery with 50% probability, independently of each other. Participants were shown the outcomes for themselves and for their partner on each trial and reported their momentary happiness every few trials. As expected, participant happiness decreased following both low lottery outcomes for themselves and for the partner. Crucially, happiness decreases following low outcomes for the partner were larger when the participant rather than their partner had made the choice, which fits an operational definition of guilt. This guilt effect was associated with BOLD signal increase in the left anterior insula. Connectivity between this region and the right inferior frontal gyrus varied depending on choice and experimental condition, suggesting that this part of prefrontal cortex is sensitive to guilt-related information during social choices. Variations in happiness were well explained by computational models based on participants' and partners' rewards and reward prediction errors. A model-based analysis revealed a left superior temporal sulcus cluster that tracked partner reward prediction errors that followed participant choices. Our findings identify neural mechanisms of guilt and social responsibility during social decisions under risk.

## Introduction

Imagine that you go out to dinner with a friend and it is your turn to choose the restaurant. You can choose between two restaurants: one that you both know well, with very predictable food of good quality, and a new restaurant that neither you nor your friend has eaten in before. You decide to try the new one. Unfortunately, while your dish is nice, your friend's turns out to be worse than the known restaurant's dishes. How would you feel? Would you feel differently if it had been your friend's turn to choose the restaurant? Being responsible for such suboptimal outcomes for others can induce a feeling of interpersonal guilt, formally described as a negative emotional response to harming someone with whom one has a positive social bond (*Baumeister, 1998*; *Baumeister et al., 1994*; *Berndsen et al., 2004*; *Tangney et al., 2007*; *Zeelenberg and Breugelmans, 2008*). Guilt influences decisions (e.g., *Charness and Dufwenberg, 2006*), and abnormal sensitivity to guilt is associated with severe social dysfunctions ranging from psychopathy to depression and anxiety, depending on whether sensitivity to guilt is, respectively, reduced or increased (*Tangney et al., 2007*).

Several brain regions have been associated with the feeling of guilt: the anterior insula (aIns) (*Bastin et al., 2016*; *Lamm and Singer, 2010*; *Piretti et al., 2023*), the dorsal cingulate cortex (*Bastin et al., 2016*; *Gifuni et al., 2017*) and the left temporo-parietal junction (*Bastin et al., 2016*; *Piretti et al., 2023*), the ventromedial prefrontal cortex (*Krajbich et al., 2009*), and with various other regions and networks involved depending on the method used to induce guilt (*Bastin et al., 2016*; *Gifuni et al., 2017*). However, we are interested in the mechanisms involved in a specific, as yet understudied aspect of guilt: the kind that can result from everyday choices that expose others to an uncertain outcome. To study this, we built on findings from experimental social decision-making tasks.

Several studies have used games from behavioural economics or perceptual tasks linked to punishment of a partner. For example, participants behaving in accord with an economic definition of guilt aversion during a trust game showed activation in insula, supplementary motor area, dorsolateral prefrontal cortex (dlPFC) and temporal parietal junction (*Chang et al., 2011*). In another study, partners received painful stimuli when participants made errors during a difficult perception task. These errors evoked activations in the left aIns and dlPFC in the participants (*Koban et al., 2013*). In a variation of this task, participants could decide to bear a proportion of their partner's pain (*Yu et al., 2014*). The level of pain taken, indicative of guilt, and activations in anterior middle cingulate cortex and aIns were higher when the pain followed errors made only by the participant rather than by both players. A multivariate reanalysis of these two datasets revealed a neural signature for guilt (*Yu et al., 2020*), with key regions including the anterior medial cingulate cortex, insula, inferior frontal gyrus (IFG), inferior temporal cortex, thalamus, and cerebellum.

One recent study paired the aforementioned perception-and-pain paradigm with dictator game decisions that allowed the participant to compensate for the partner's outcome (*Gao et al., 2018*). The guilt context increased advantageous-inequity aversion and decreased disadvantageous-inequity aversion and affected the neural correlates of these inequities (respectively, mentalizing-related and emotion- and conflict-related regions). Finally, another study contrasted guilt and shame: confederates either experienced economic loss due to bad participant advice (participants experienced guilt), or experienced no loss when participants' bad advice was correctly refused by the confederate (participants experienced shame) (*Zhu et al., 2019*). Guilt relative to shame activated supramarginal gyrus and temporo-parietal junction as well as orbitofrontal cortex, ventrolateral, and dorsolateral prefrontal cortex. Multivariate analyses revealed that guilt could be distinguished from shame based on activation in ventral anterior cingulate cortex and dorsomedial prefrontal cortex.

One landmark study reported on the mechanisms involved in seeking or avoiding responsibility in decisions affecting a group of individuals, and reported involvement of medial prefrontal cortex, aIns and temporo-parietal junction (*Edelson et al., 2018*). In this elegant study, participants could delegate their choice between a risky and a safe option to a group or decide themselves; in separate conditions, the choice affected payoff either only for the participant or for all group members. Interestingly, most participants displayed responsibility aversion, and this effect could not be explained by guilt, suggesting people associate a psychological cost with assuming responsibility for others' outcomes. However, in many situations, one does not have the possibility to delegate a decision to others, and choosing a risky option in such a case may evoke guilt when the outcome for others is negative. The neural mechanisms underlying guilt evoked during such situations of social responsibility are still unknown. These questions are the subject of the present study. Similarly to Edelson et

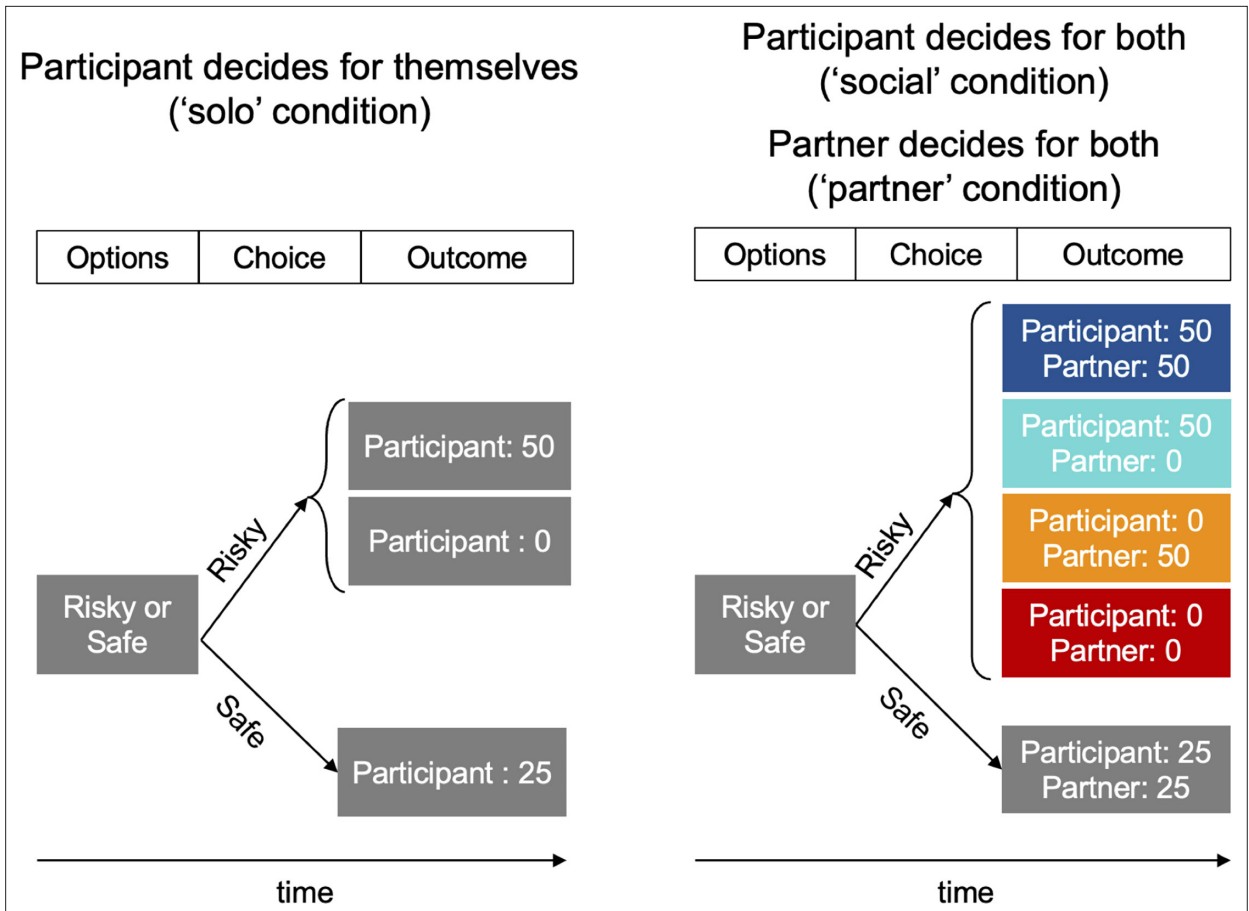

**Figure 1.** Experimental design. In every trial, participants were presented with pairs of monetary options (a safe and a risky option; the risky option was a lottery with equally probable high and low outcomes). There were three conditions: a non-social 'Solo' condition, in which the participant's choice led to an outcome just for themselves (left panel); and two conditions in which choices were made by the participant ('Social' condition) or by their partner ('Partner' condition) and led to outcomes affecting both players (right panel). Importantly, selecting the risky option in the social or partner conditions led to the lottery being played out independently for both players; that is, participant and partner could receive the high or low outcome independently from each other (coloured boxes). Selecting the safe option led to both players receiving an equal outcome.

al.'s and more recent studies (*Arioli et al., 2023*; *Fareri et al., 2022*), our paradigm leveraged a risky choice task with social conditions.

In our experiment, participants were paired with a partner and played an 'ice-breaker' game that created a positive social bond between them, increasing the likelihood of feeling empathy and guilt for each other (*Baumeister, 1998*; *Julle-Danière et al., 2020*; *Loewenstein et al., 1989*). Participants or their partner then chose between risky and safe monetary options in three conditions (*Figure 1*). In the *Social* condition, the outcome of choices affected both participant and partner; participants were and felt responsible for these outcomes because they had agency over their decisions and knew that they could have chosen otherwise (*Frith, 2014*). We contrasted this condition with similar choices made by a simple expected-value-maximizing algorithm posing for the partner (*Partner* condition), and participant choices only affecting the participant themselves (*Solo* condition, acting as control). Importantly, we assessed the emotional impact of the outcomes of these choices by monitoring participants' happiness every two trials (*Rutledge et al., 2014*; *Rutledge et al., 2016*). This allowed us to fit computational models to happiness data and search for networks sensitive to reward prediction errors resulting from participant or partner choices. We ran two experiments, Study 1 outside the MRI scanner and Study 2 during fMRI, with separate groups of participants.

We analysed our behavioural data using several complementary methods: choices were modelled with mixed-effects regressions serving as manipulation checks; risk preferences expressed in choices were assessed using a comprehensive expected utility model as well as with a simpler, more robust

'risk premium' approach; and happiness data were fitted, in addition to the computational models, with several linear mixed models (LMMs) to assess the impact of both the participant's and their partner's rewards, the impact of agency and their interactions. Inspired by findings reported in previous neuroimaging of social emotions, we also used several methods to analyse our fMRI data, including conventional methods (both region-of-interest and mass univariate); mixed-effects regression models; computational model-based analyses (inspired by, e.g., *Konovalov et al., 2021a*; *Rutledge et al., 2014*); and functional connectivity (e.g., *Edelson et al., 2018*; *Konovalov et al., 2021a*). The behavioural modelling is thus complemented by neuroimaging analyses that offer insight about both the activity in regions associated with guilt as well as their place in a wider network, providing an in-depth comprehensive analysis of the mechanisms behind guilt evoked by social responsibility.

## Results
### Behaviour
#### Overview

We first verified that participants' choices were reasonable in the *Solo* and *Social* conditions, then assessed whether these choices and the risk preferences they revealed changed depending on whether participants chose just for themselves (*solo*) or for themselves and the interaction partner (*social*). Next, we assessed whether momentary happiness varied with decision outcomes and whether responsibility influenced this relationship. Lastly, we fitted computational models to the happiness data. While Study 1 (behaviour only) was run before Study 2 (fMRI), we will report the results of both studies together as their results were highly consistent.

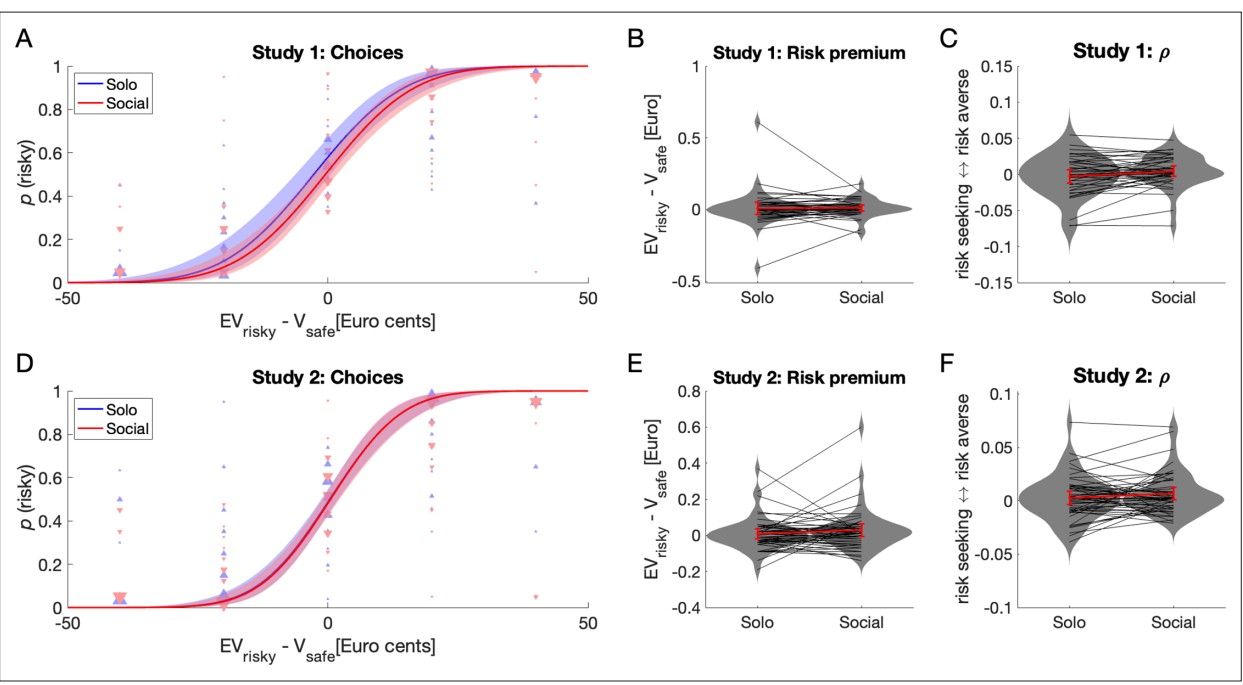

**Figure 2.** Participant choices in Studies 1 (outside fMRI, *N* = 40) and 2 (inside fMRI, *N* = 44). (**A, D**) The probability of choosing the risky option (lottery) in both Solo and Social conditions is well explained by the difference in expected value of the risky and safe choice options (EV$_{risky}$ − V$_{safe}$). Participants chose the risky option slightly more often in the Solo condition than in the Social condition in Study 1 (**A**) but not in Study 2 (**D**). Lines are predicted values of a logit linear mixed regression model fitted to the choice data (see Results). Error areas indicate 95% pointwise confidence intervals for the predicted values. Triangles indicate individual average choice proportions binned by EV$_{risky}$ − V$_{safe}$ value; size of triangle reflects the number of participants contributing to a datapoint. Blue upward-pointing and red downward-pointing triangles are data from the Solo and Social conditions, respectively. (**B, E**) Risk premiums did not differ between Solo and Social conditions. (**C, F**) Values of the risk aversion parameter $\rho$ in the Solo and Social conditions were broadly consistent with Risk premium values, but showed that participants were slightly more risk averse in the Social than in the Solo condition in Study 1 only (see Results). In panels B, C, E and F, grey lines and markers show individual data, red lines show means with 95% confidence intervals about the means, and grey areas are kernel density plots representing the distribution of the data.

## Choices: manipulation check

As expected, participants' probability of choosing the risky option (lottery) increased with the difference between the expected value of the lottery and the value of the safe option (Study 1: *Figure 2A*, $t(4796) = 9.26$, $p < 3.1e^{-20}$, $\beta = 0.074$, 95% CI = [0.059 0.090]; Study 2: *Figure 2D*, $t(3829) = 10.62$, $p < 5.3e^{-26}$, $\beta = 0.093$, 95% CI = [0.075 0.110]; mixed-effects regressions, see *Equation 1*, detailed results are reported in *Appendix 1—table 1*). Participants chose the lottery more often in the *Solo* condition than in the *Social* condition in Study 1 ($t(4796) = 2.54$, $p = 0.011$, $\beta = 0.164$, 95% CI = [0.038 0.291]), but this difference was not found in Study 2 ($t(3829) = 0.23$, $p = 0.82$, $\beta = 0.015$, 95% CI = [–0.109 0.138]). There was no significant interaction between the difference in expected values and experimental conditions in either study ($p > 0.52$).

## Choices: risk preferences

To better assess whether people's risk preferences varied between the *Solo* and the *Social* condition, we evaluated two additional measures. First, we calculated for each participant a 'risk premium', defined as the difference between the expected value of a lottery and its certainty equivalent (see *Equation 2* in Methods; positive risk premiums indicate risk aversion). Second, we used an expected utility theory (EUT) approach to calculate a parameter $\rho$ that describes a decision-maker's risk attitude under the assumption of constant absolute risk aversion (see *Equation 3* in Methods). We used two measures because fitting the EUT model, the more comprehensive measure that takes into account all choices of a participant, requires many trials to be fitted reliably, a condition that was not satisfied in many participants of both studies; in contrast, the risk premium's single-point measure of risk aversion could be estimated in all participants. Risk premiums did not differ between *Social* and *Solo* conditions (Study 1: *Figure 2B*, $t(39) = 1.53$, $p = 0.134$, Cohen's $d = 0.24$, $BF_{10} = 0.49$; Study 2: *Figure 2E*, $t(43) = –0.21$, $p = 0.84$, $d = –0.03$, $BF_{10} = 0.17$). Turning to the EUT approach, as differences in risk attitudes for gains and losses are well known (*Kahneman and Tversky, 1979*), we first estimated $\rho$ separately for gain and loss trials. As the number of these types of trials varied across participants, we only obtained reliable estimates for both types of trials in some participants (Study 1: 18 of 40; Study 2: 16 of 44). As $\rho$ did not vary between gain and loss trials (Study 1: $t(17) = 0.21$, $p = 0.84$, $d = 0.05$; Study 2: $t(15) = –0.61$, $p = 0.55$, $d = 0.15$; paired $t$-test), we then pooled across gain and loss trials, then estimated and compared $\rho$ for the *Solo* and *Social* conditions using paired $t$-tests. We found that participants were slightly more risk averse in the *Social* than in the *Solo* condition in Study 1 (*Figure 2C*, $t(39) = 2.27$, $p = 0.03$, $d = 0.36$, $BF_{10} = 1.69$) but not in Study 2Study 2 (*Figure 2F*, $t(43) = 1.40$, $p = 0.17$, $d = 0.21$, $BF_{10} = 0.41$). Risk premium and $\rho$ were highly consistent with each other across participants in both conditions for both studies (Study 1: *Solo* condition: $F(1,38) = 69.9$, $p < 0.001$, $R^2 = 0.65$; *Social* condition: $F(1,38) = 42.0$, $p < 0.001$, $R^2 = 0.55$; Study 2: *Solo* condition: $F(1,41) = 57.9$, $p < 0.001$, $R^2 = 0.59$; *Social* condition: $F(1,41) = 74.2$, $p < 0.001$, $R^2 = 0.64$; these and all subsequent regressions are robust).

In sum, participants showed very similar risk preferences when making decisions affecting only themselves (*Solo* condition) or themselves and their partner (*Social* condition), with a tendency towards higher risk aversion in the *Social* condition in Study 1.

## Momentary happiness: links to reward

Momentary happiness was assessed every two trials in a similar manner to previous studies by *Rutledge et al., 2014*; *Rutledge et al., 2016*. Across all trials, in both studies, participant momentary happiness correlated with rewards obtained in the current trial by the participant and by the partner (Correlations with participant reward, Study 1: $F(1,3598) = 691.5$, $p < 0.001$, $R^2 = 0.16$; partner reward, Study 1: $F(1,2426) = 128.6$, $p < 0.001$, $R^2 = 0.05$; participant reward, Study 2: $F(1,2630) = 650.8$, $p < 0.001$, $R^2 = 0.20$; partner reward, Study 2: $F(1,1735) = 111.8$, $p < 0.001$, $R^2 = 0.06$; *Figure 3A, B, E, F*). Based on previous findings (*Rutledge et al., 2016*), we reasoned that participants' momentary happiness would be influenced not only by rewards obtained in the current trial, but also by expected rewards, reward prediction errors, rewards received in previous trials, and differences between rewards obtained by the participant and the partner. Crucial for our research question, we aimed to assess whether responsibility for these rewards, that is, taking into account whether the rewards occurred following choices made by the participant or the partner, would also influence variations in happiness.

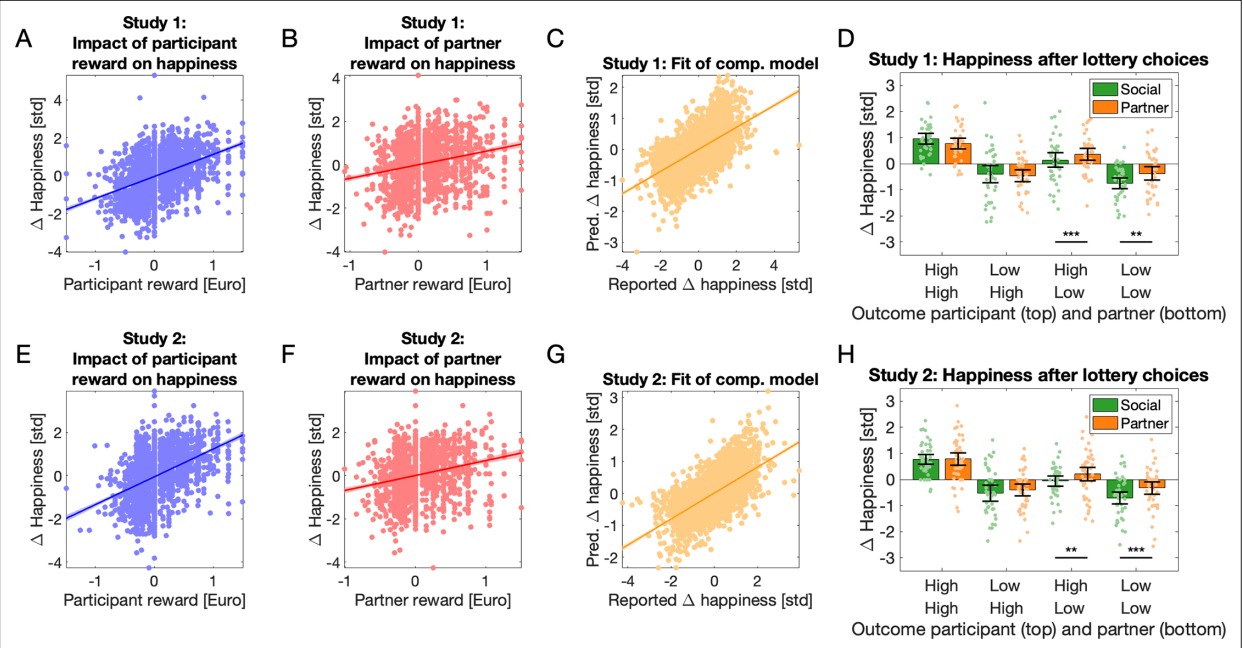

**Figure 3.** Participant momentary happiness in Studies 1 (**A–D**) and 2 (**E–H**). Happiness varied with rewards received by the participant (**A, E**) and by the partner (**B, F**). Each dot is one trial; data are pooled across participants. Lines are fitted regression lines. A computational model taking into account expected, previous and current rewards, reward prediction errors for both participant and partner, and decision-maker (Responsibility Redux model, see Results) predicted the variations in participants' momentary happiness well (**C, G**, and *Table 1*). Changes in momentary happiness after lottery choices in Social and Partner conditions varied with lottery outcome and decision-maker (**D, H**). Data were binned according to outcome for each participant and decision-maker (Social = participant chose the lottery, Partner = partner chose the lottery). Crucially, responsibility for low lottery outcomes for the partner decreased participant happiness more than the same outcomes following partner choices (see Results), which fits the definition of interpersonal guilt. In D and H, dots are individual datapoints, the bar indicates the mean, the error bars are 95% confidence intervals about the mean, and the stars indicate the significance of the 'guilt effect' (see text): ***p < 0.001; **p < 0.01.

The online version of this article includes the following figure supplement(s) for figure 3:

**Figure supplement 1.** Parameter recovery for Responsibility Redux model.

Following Rutledge and colleagues' methodology, which considers that changes in momentary happiness in response to outcomes of a probabilistic reward task are explained by the combined influence of recent reward expectations and prediction errors arising from those expectations, we fitted computational models to each participant's happiness data. In these models, certain rewards, the expected value of chosen lotteries, reward prediction errors and additional outcome parameters are modelled separately with influences that decay exponentially with time (see Methods for equations and details). We first fitted a '*Basic*' model (*Equation 4*) containing regressors coding for certain reward, expected value of chosen lotteries and participant reward prediction errors to evaluate whether these fundamental reward variables explained variations in momentary happiness. This model explained the data reasonably well (see *Table 1*). Next, based on the notion that inequality between rewards received by the participant and the partner would influence happiness (e.g., *Loewenstein et al., 1989*), we assessed the fit of an '*Inequality*' model that included one term modelling the difference between the participant and partner outcomes (*Equation 5*). We also assessed the fit of *Rutledge et al., 2016* '*Guilt-envy*' model in which advantageous and disadvantageous inequality were modelled separately (*Equation 6*). Next, we tested a '*Responsibility*' model in which separate terms represented the partner's reward prediction errors resulting from choices made by the participant and by the partner (*Equation 7*). Finally, we optimized the Responsibility model by removing the regressor modelling partner reward prediction errors resulting from choices made by the partner, yielding a '*Responsibility Redux*' model (*Equation 8*).

All models explained variations in happiness reasonably well (*Table 1*). Overall, the *Responsibility* and *Responsibility Redux* (*Figure 3C, G*) models explained the data best (highest $R^2$/adjusted $R^2$ or lowest AIC/BIC); a likelihood ratio test (*Equation 9*) revealed that the *Responsibility* model fitted

**Table 1.** Fits of computational models to momentary happiness data.

| Model | N param | Mean $R^2$ | Mean $R^2$adj | BIC | AIC |
|---|---|---|---|---|---|
| Study 1 | | | | | |
| Basic | 3 | 0.328 | 0.305 | **−1000** | −1399 |
| Inequality | 4 | 0.346 | 0.316 | −916 | −1416 |
| Guilt-envy | 5 | 0.354 | 0.316 | −785 | −1385 |
| Responsibility | 5 | **0.370** | **0.333** | −866 | −1466 |
| Responsibility Redux | 4 | 0.361 | 0.331 | −999 | **−1499** |
| Study 2 | | | | | |
| Basic | 3 | 0.374 | 0.340 | −693 | −1062 |
| Inequality | 4 | 0.394 | 0.350 | −620 | −1080 |
| Guilt-envy | 5 | 0.405 | 0.350 | −491 | −1043 |
| Responsibility | 5 | **0.433** | **0.380** | −616 | −1168 |
| Responsibility Redux | 4 | 0.422 | 0.379 | **−735** | **−1195** |

BIC, Bayesian Information Criterion; AIC, Akaike's Information Criterion. BIC and AIC values are summed across participants. As in previous work by **Rutledge et al., 2014**; **Rutledge et al., 2016**, model fits were performed with individually Z-scored happiness ratings. Best values of each variable for each study are highlighted in bold font. For model details, see Results and Methods (**Equations 4–8**).

better than all the other models, including the *Responsibility Redux* model (Study 1: all LR ≥47.36, p < 0.0001; Study 2: all LR ≥77.83, p < 0.0001). We also compared the $R^2$ and adjusted $R^2$ values obtained for each participant and model using *t*-tests, for both studies (average values are reported in **Table 1**). The *Responsibility* model yielded higher $R^2$ values than all the other models (Study 1: all *t* > 3.6, p < 0.007; Study 2: all *t* > 2.9, p < 0.034; Bonferroni-corrected *t*-tests) except for the *Guilt-envy* model in the data of Study 1 (*t* = 2.19, p = 0.17). The *Responsibility* and *Responsibility Redux* models yielded higher *adjusted $R^2$* than the *Basic* model (Study 1: all *t* > 3.6, p < 0.01; Study 2: all *t* > 3.6, p < 0.01). Weights for certain rewards, expected value of the lotteries, participant reward prediction errors and the forgetting factor *γ* were overall positive across participants, models and studies (Study 1: all *Z* > 5.4, p < 0.0001; Study 2: all *Z* > 4.9, p < 0.0001, Wilcoxon sign-rank tests were used because data were not normally distributed). Medians of the forgetting factor *γ* varied little across models, ranging from 0.39 to 0.44 in Study 1 and from 0.42 to 0.46 in Study 2, indicating stable influences of previous trials on happiness across models. Participants' reward prediction errors (sRPE) influenced happiness more than partner's reward prediction errors, whether the latter resulted from the participant or the partner's choices (respectively, *social_pRPE* and *partner_pRPE*): weights for *sRPE* were higher than for *social_pRPE* or *partner_pRPE* (Study 1: all *Z* > 6.0, p < 0.001; Study 2: all *Z* > 3.7, p < 0.003). The stability of these estimated parameters was verified using a parameter recovery procedure (see Methods and **Figure 3—figure supplement 1**). These results thus replicate the finding that outcomes of risky social decisions influence momentary happiness (**Rutledge et al., 2016**). Crucially, we find here that the partner's reward prediction errors (*social_pRPE* and *partner_pRPE*) contributed to explaining changes in participants' momentary happiness: the *Responsibility* and *ResponsibilityRedux* models explained the data better than the models without these parameters (see **Table 1**). In particular, the partner's reward prediction errors resulting from the participants' decisions (*social_pRPE*), that is, those pRPE for which participants were responsible contributed to explaining our data (weights for *social_pRPE* were greater than 0: *Responsibility model*: Study 1: *Z* = 2.85, p = 0.004, Study 2: *Z* = 3.26, p = 0.001; *ResponsibilityRedux model*: Study 1: *Z* = 2.93, p = 0.003, Study 2: *Z* = 3.30, p = 0.001; weights for *social_pRPE* tended to be higher than weights for *partner_pRPE*: *Responsibility model*: Study 1: *Z* = 2.14, p = 0.033; Study 2: *Z* = 1.41, p = 0.16).

## Momentary happiness: effects of agency, responsibility, and guilt

Next, we assessed whether happiness varied depending on the participant's agency (*Social + Solo* vs. *Partner*), and found happiness to be lower when the participant chose, independent of the outcome

(Study 1: $t(3600)$ = –3.92, p < 0.0001, $\beta$ = –0.14, 95% CI = [−0.20 to 0.07]; Study 2: $t(2870)$ = –6.07, p < 0.0001, $\beta$ = –0.24, 95% CI = [−0.31 to 0.16]). This is interesting in itself and may reflect the drive behind responsibility aversion reported by Edelson et al.'s 2018 study: being assigned the role of the decider in a social setting may make people slightly unhappy, perhaps due to 'weight of the responsibility'. To specifically search for a sign of interpersonal guilt, we analysed happiness values reported after outcomes of lottery choices in the *Social* and *Partner* conditions using conventional LMMs (*Equation 10*). Fitting several models revealed that lottery outcomes influenced happiness, and that the identity of the decision-maker played a role (*Appendix 1—table 2*). Crucially, the interaction between partner outcome and decision-maker was significant (Study 1: $t(1180)$ = 3.52, p = 0.0004, $\beta$ = 0.37, 95% CI = [0.16 0.58]; Study 2: $t(937)$ = 2.85, p = 0.0045, $\beta$ = 0.33, 95% CI = [0.10 0.56]). When the partner received the low lottery outcome, participant happiness was lower when they rather than the partner had chosen the lottery (*Figure 3D, H*). When the partner received the low lottery outcome of a participant-chosen lottery, they would presumably feel let down because the participant could have chosen the better safe option. As participants knew this, they were likely to feel 'simple guilt' (*Battigalli and Dufwenberg, 2007*). This behavioural effect (difference in happiness obtained when the partner received low lottery outcomes after participant rather than partner choices) is thus compatible with 'simple guilt', and we will thus refer to it as 'guilt effect'. The 'guilt effect' occurred whether the participant received the high lottery outcome (Study 1: $t(39)$ = –3.58, p < 0.001, $d$ = 0.56, $BF_{10}$ = 32; Study 2: $t(43)$ = –2.68, p = 0.01, $d$ = 0.4, $BF_{10}$ = 3.8) or the low lottery outcome (Study 1: $t(39)$ = –3.39, p = 0.002, $d$ = 0.54, $BF_{10}$ = 19; Study 2: $t(43)$ = –3.58, p < 0.001, $d$ = 0.54, $BF_{10}$ = 33.5). In Study 2, we will compare individual 'guilt effect' values to individual guilt-related brain activation patterns (see end of the section on BOLD signal results). Responsibility for choices did not influence happiness following positive lottery outcomes for the partner (both studies, all $|t|$ < 1.3, p > 0.2, $BF_{10}$ < 0.2).

In sum, in both studies, we found evidence that participants' happiness was influenced by the outcomes of choices for their partner, especially by outcomes resulting from participant decisions, that is, partner outcomes for which participants were responsible. Within these outcomes, participants felt worse following low lottery outcomes for the partner if those outcomes were consequences of their own choice rather than the partner's, which we interpret as interpersonal guilt.

## BOLD signal

Next, we sought to uncover the neural mechanisms associated with our guilt effect and those involved in tracking consequences of participants' decisions on their partner. We analysed the BOLD responses of brain regions engaged during decision-making and at the time of receiving the outcomes of the choice using conventional as well as computational model-based analyses, using the fMRI data collected in Study 2.

### Brain regions engaged during social decision-making

Given that our task involved several conditions and successive trial components, we aimed to first replicate previous results related to the neural correlates of decisions under economic risk. We searched for brain regions engaged more when participants chose the risky instead of the safe option and found such responses in the bilateral ventral striatum (Cohen's $d$ = 0.72 and 0.85 in the left and right clusters, respectively; *Figure 4A* and *Appendix 1—table 3*), which replicates previous findings (*Cui et al., 2022*; *Preuschoff et al., 2006*). Next, we aimed to identify the brain regions associated with social decision-making under risk and searched for regions more engaged during decisions in the *Social* condition compared to the *Solo* condition. Three significant clusters of voxels were identified (*Figure 4B* and *Appendix 1—table 3*), in the precuneus ($d$ = 0.79), the left temporo-parietal junction (TPJ; $d$ = 0.59) and the medial prefrontal cortex (mPFC; $d$ = 0.54). This also replicates previous findings associating this region with social decisions (e.g., *Fareri et al., 2012*; *Jung et al., 2013*; *Nicolle et al., 2012*; *Ogawa et al., 2018*; *Piva et al., 2019*).

To assess whether activations in these five regions were sensitive to the interaction between choice and condition using a sensitive method, we analysed the responses in all the voxels in these regions using LMMs. Our model included factors *Choice* (*Safe* or *Risky*), *Condition* (*Social*, *Partner* or *Solo*), *Run* (run one or two), and *ROI* (left and right ventral striatum, mPFC, precuneus, and TPJ), with all interactions. We also tested the same model without the factor *Run*, but this fitted less well (higher BIC) and was thus discarded. As all factors and interactions were significant (data not shown for

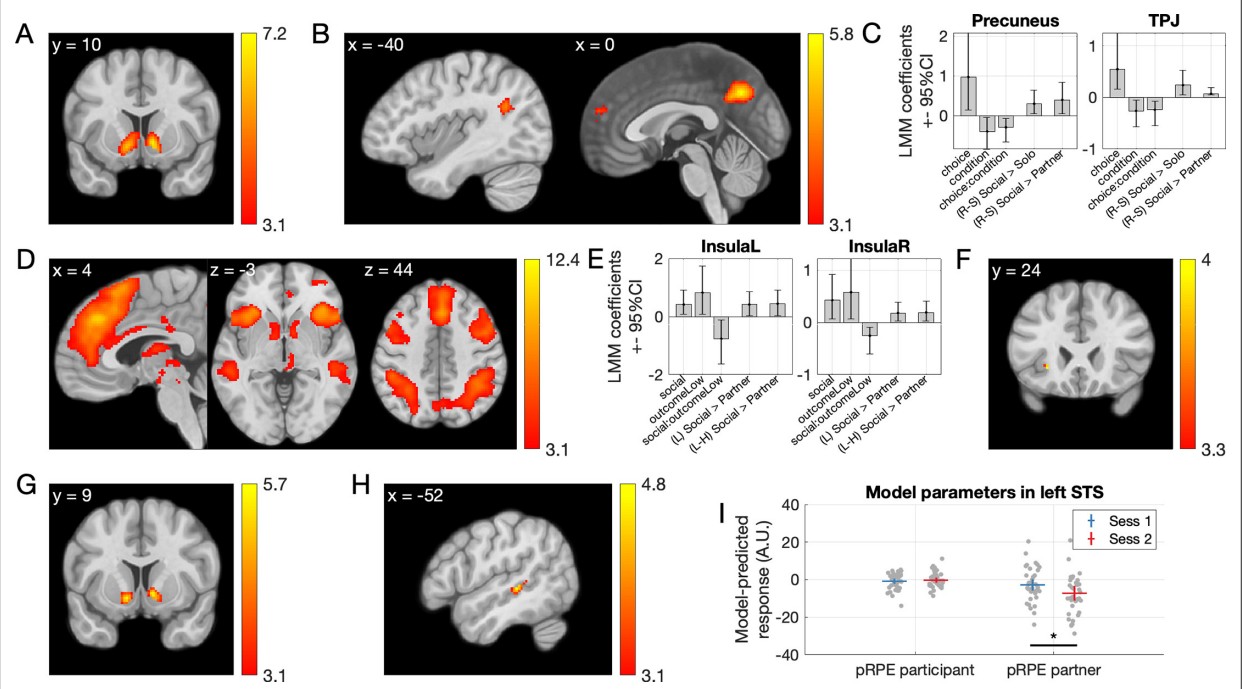

**Figure 4.** BOLD responses. (**A**) Regions showing a greater response when participants chose the risky (lottery) rather than the safe option, irrespective of Social or Solo condition. (**B**) Regions showing a greater response when participants chose for both themselves and their partner rather than just for themselves (Social > Solo). (**C**) Coefficients of linear mixed models (LMMs) indicate that two of these regions, precuneus and TPJ, were most active when participants chose the lottery in the Social condition. R–S indicates results of LMMs based on the Risky–Safe response difference. All coefficients and differences reported are significantly different from 0 (see *Appendix 1—table 4*). (**D**) Brain regions more active during receipt of the outcomes of lotteries than safe choices (all conditions). (**E**) Coefficients of LMMs indicated that insula ROIs (see D) mirrored the guilt effect observed in our behavioural data: voxels here responded more to low lottery outcomes (L) for the partner when these resulted from participant's rather than the partner's choices, even when responses to high outcomes were subtracted (L–H). All coefficients and differences reported are significantly different from 0 (see *Appendix 1—table 6*). (**F**) A mass-univariate, voxel-wise analysis showed a compatible result: A cluster of voxels within the left insula ROI showed higher responses to low lottery outcomes for the partner if these resulted from participant rather than partner choices. (**G**) Activation in bilateral ventral striatum explained by a computational model-based regressor coding participant rewards. (**H**) Within brain regions sensitive to outcomes of risky choices, one cluster in the left superior temporal sulcus region showed a higher response to partner reward prediction errors resulting from participant rather than partner choices. (**I**) Response in this cluster to the computational-model-based regressors coding participant reward prediction resulting from participant and partner choices, for both sessions of the experiment. All results shown survive thresholding at p < 0.05 corrected for multiple comparisons at the cluster level, based on a voxel-wise uncorrected threshold of p < 0.001. Colours in panels A–I indicate T values. In C and E, bars indicate the estimated coefficients. In C, E, and I, error bars are 95% confidence intervals.

sake of brevity), we applied the same models to each region separately (*Appendix 1—table 4*). All regions showed a significant interaction between *Condition* and *Choice*, prompting us to examine these responses in detail. As we were particularly interested in the response to *Risky* decisions, we subtracted from it the responses to *Safe* decisions (labelled as '(R–S)' in *Figure 4C*) and compared this difference between (1) *Social* and *Solo* conditions, and (2) between the *Social* and *Partner* conditions using LMMs. As above, we tested these models both with and without the factor *Run* and associated interaction, and we report the best-fitting model (see *Appendix 1—table 5* and *Appendix 1—table 6*). Only the precuneus and TPJ showed positive differences in both comparisons (*Figure 4C*), indicating that these regions were most active when participants chose the lottery in the *Social* condition, the critical situation in which participants assume responsibility over others.

## Brain regions engaged during receipt of outcomes
Next, we focused on responses during choice outcomes. A cluster of voxels more active during receipt of lottery outcomes than outcomes of safe choices was identified in the bilateral anterior insula, dorsal mPFC (dmPFC), right superior temporal sulcus (STS), bilateral ventral striatum, right dorsolateral prefrontal cortex, and bilateral inferior parietal lobe (*Figure 4D*; for details including effect sizes, see

*Appendix 1—table 7*). Except for the STS, all these regions have been previously associated with processing of risk and/or ambiguity (*Wu et al., 2021*).

In our definition, guilt occurs due to responsibility for low lottery outcomes for the partner. To search for regions involved in this situation, we again analysed voxel responses using LMMs. Our first model included factors *Lottery outcome* (*High* or *Low*), *Condition* (*Social* or *Partner*), *Run* (run one or two), and *ROI* (left and right ventral striatum, left and right insula, left and right parietal cortex, dmPFC, right prefrontal, and right middle temporal regions), with all interactions. We also tested the same model without the factor *Run*, which fitted less well and was discarded. As in the analysis of responses to decisions above, all factors and interactions were significant (data not shown for sake of brevity), and we applied the same models to each region separately (*Appendix 1—table 8*). To identify regions likely to be involved in the guilt effect, we selected those satisfying two conditions: higher activity in the *Social* compared to the *Partner* condition, and a significant *Social:LowOutcome* interaction. This procedure revealed the insulae (*Figure 4E*) and the right middle temporal cortex (trend significant *Social* vs. *Partner* difference in the latter region). To test if these regions responded more to *LowOutcomes* in the *Social* condition, we ran additional models on the responses to *Low lottery outcomes* only (labelled as '(L) Social > Partner' in *Figure 4E*), and on this response minus the response to *High lottery outcomes* (labelled as '(L–H) Social > Partner' in *Figure 4E*); and indeed, the insulae showed the sought-after effect (see *Appendix 1—table 9*, *Appendix 1—table 10*, and *Figure 4E*). Thus, activation in our insula ROIs increased in situations during which participants experienced guilt for low outcomes impacting their partner, compared to similar outcomes resulting from the partner's choices.

To attempt to confirm these results with a classic fine-grained mass-univariate voxel-wise analysis, we searched, within regions responding more to outcomes of risky compared to safe choices, for higher responses to low lottery outcomes for the partner following participant choices compared to the same outcomes resulting from partner choices. We found a weak response in a small cluster within the left anterior insula (peak $T = 3.95$, $d = 0.59$, 22 voxels, peak intensity at $[-28\ 24\ -4]$; *Figure 4F*). Given the documented association between anterior insula and guilt (see Introduction), we proceeded to test whether this result survived correction for family-wise errors due to multiple comparisons restricted to the left anterior insula grey matter [defined anatomically and thus independently from our findings, as the anterior short gyrus, middle short gyrus, and anterior inferior cortex in an anatomical maximum probability map (*Faillenot et al., 2017*)]. This correction resulted in a p value of 0.024. This result, although it is only a small effect in a small cluster, is consistent with the mixed model analysis reported earlier.

## Neural correlates of responsibility for partner reward prediction errors revealed by computational model-based analysis

Next, we attempted to explain BOLD responses using predictions of the '*Responsibility*' computational model (see Behaviour/Computational modelling of happiness data, above). We used the model to create expected BOLD responses for each participant (see Methods) and as a manipulation check searched for responses in ventral striatum evoked by participant rewards (*O'Doherty et al., 2004*; *O'Doherty et al., 2007*). We found that activation in bilateral ventral striatum indeed increased with the amount of expected certain rewards and the expected values of chosen lotteries (left: $p_{FWE} = 0.002$, $T = 5.63$, $d = 0.75$, $Z = 5.41$, 110 voxels, peak at MNI $[-14\ 8\ -8]$, right: $p_{FWE} = 0.005$, $T = 5.46$, $d = 0.70$, $Z = 5.26$, 80 voxels, peak at MNI $[10\ 10\ -4]$, correction for multiple tests applied across the whole brain; *Figure 4G*). We thus used this model to search for voxels responding more to partner reward prediction errors resulting from participant rather than partner choices, within the regions sensitive to outcomes of risky choices. We found this effect in one cluster within the left STS ($p_{FWE} = 0.022$, $T = 4.70$, $d = 0.53$, $Z = 4.57$, 100 voxels, peak at MNI $[-52\ -32\ 0]$; *Figure 4H*). Responses to the computational-model-based regressors coding for partner reward prediction errors resulting from participant and from partner choices in this cluster are shown in *Figure 4I*. This finding suggests that this region of the left STS tracks a partner's unexpected outcomes less when they do not follow from the participant's decisions.

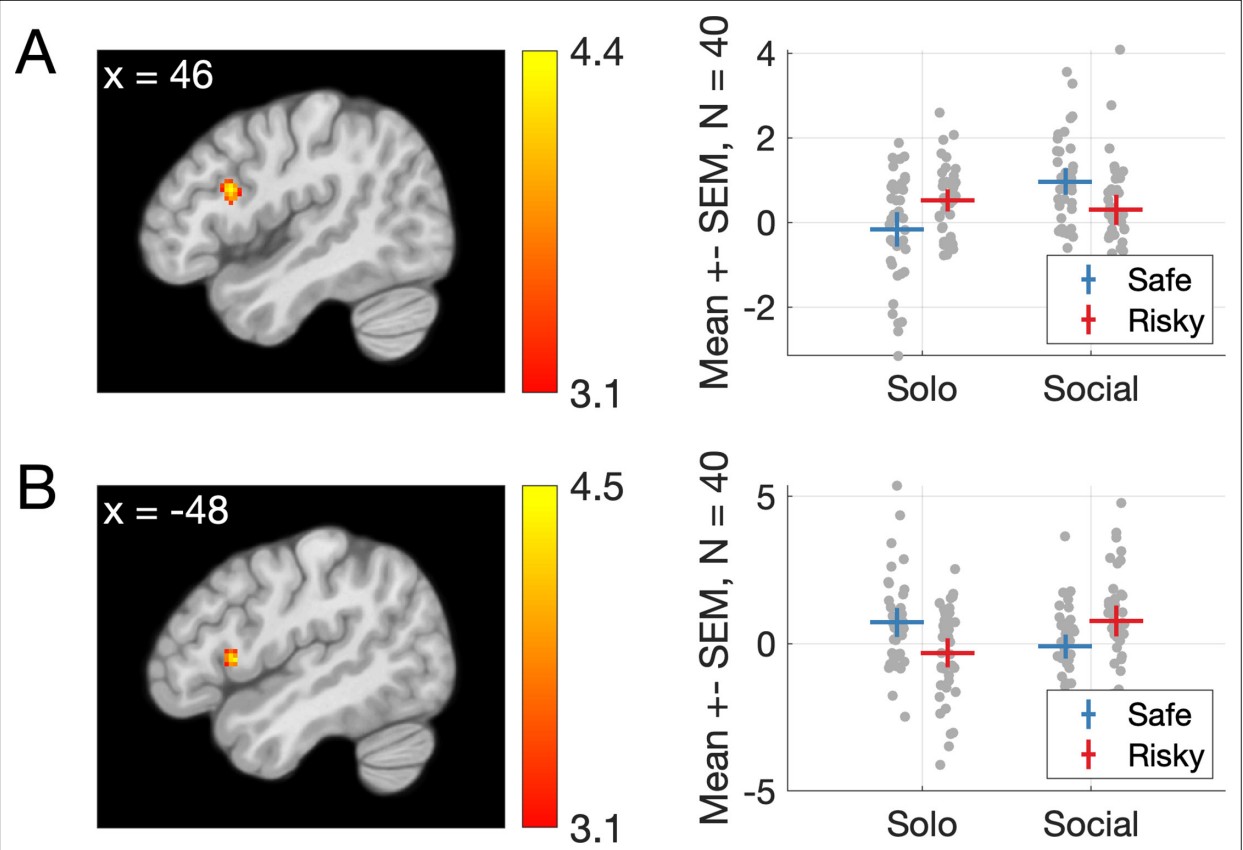

**Figure 5.** Changes in functional connectivity between the left anterior insula (seed) and a cluster in the right inferior frontal gyrus at the time of the choice as a function of condition (Social vs. Solo) and choice (Risky or Safe). In the righthand panel, dots are data of individual participants, the markers represent means, and error bars indicate 95 confidence intervals about the mean.

The online version of this article includes the following figure supplement(s) for figure 5:

**Figure supplement 1.** Functional connectivity with the left TD-model-defined superior temporal sulcus (STS; seed) during choices in Solo and Social conditions.

## Functional connectivity

Functional connectivity analyses have revealed differences in networks engaged by social and self-only choices (*Jung et al., 2013*; *Ogawa et al., 2018*), interactions between midbrain and anterior cingulate during compensation for guilt (*Yu et al., 2014*), and links between insula connectivity and responsibility aversion (*Edelson et al., 2018*). We hypothesized that connectivity with regions that showed guilt- and responsibility-related responses during the outcome phase (see previous paragraph) might change depending on whether participants made decisions for themselves only or for themselves and their partner, and depending on the type of choice (*Safe* or *Risky*). To this end, we used the left insula and left STS as seed regions for whole-brain seed-to-voxel psychophysiological interaction (PPI) analyses (see Methods), to search for connectivity changes, during the choice phase of the trial, as a function of *Condition* and *Choice* (i.e., voxels with a significant interaction to *Condition* by *Choice*). The first analysis revealed a cluster in the right IFG whose connectivity to the insula (the seed region) was highest when participants made *Risky* choices for themselves and *Safe* choices for both players ($p_{FWE}$ = 0.020, $T$ = 4.34, $d$ = 0.80, $Z$ = 4.21, 115 voxels, peak at MNI [46 16 22]; *Figure 5*). A smaller cluster in the left IFG showed the same effect but did not survive correction for multiple tests ($p_{uncorrected}$ = 0.001, $T$ = 4.08, $Z$ = 3.97, 38 voxels, peak at MNI [−38 −2 26]). The second analysis revealed a smaller cluster in the left IFG that did not survive corrections for multiple tests, where connectivity with the left STS (the seed region) showed the opposite pattern: connectivity was highest when participants made *Safe* choices for themselves and *Risky* choices for both players ($p_{uncorrected}$ = 0.001, $T$ = 4.44, $Z$ = 4.30, 35 voxels, peak at MNI [−48 14 6]; *Figure 5—figure supplement 1*).

## Comparison with multivariate neural guilt signature (*Yu et al., 2020*)

A recent study by Yu and colleagues re-analysed the results of two previous neuroimaging studies of guilt and obtained a neural multivariate guilt-related brain signature (GRBS) (*Yu et al., 2020*). As the GRBS and code to compare neural responses to it is freely available at GitHub (*Wager, 2019*), we compared this brain signature to the neural responses obtained in our task (response to low partner outcomes resulting from participant vs. partner responses). The dot products between individual responses and the GRBS varied between –40.1 and 36.7, but overall these values were positive (mean = 5.22; median = 6.97; sign test: p = 0.017; *Cliff's Delta* = 0.4 = medium effect size; data are not normally distributed). We assessed whether inter-individual differences in these dot product values correlated with the behavioural guilt responses, but did not find a significant association [*Spearman's Rho* = –0.058, p = 0.725].

## Discussion

We report findings from an experiment on social responsibility and guilt in risky economic decisions, and their neural correlates. Being responsible for choosing a lottery that yielded a low outcome for a partner made our participants feel worse than witnessing the same outcome resulting from their partner's choice, which we interpret as interpersonal guilt; although we note that we have not asked participants specifically about which emotion they felt in these situations. Activation in the left anterior insula (aIns) reflected this effect, replicating previous associations between this region and feelings of guilt. Whole-brain activation patterns also resembled a neurometric marker of guilt (*Yu et al., 2020*). Connectivity between aIns and the right IFG varied depending on whether participants chose the risky or safe option and whether only the participant or both themselves and the partner were affected by the outcome of this choice, suggesting that this part of prefrontal cortex is sensitive to guilt-related information during social choices. Computational models explained trial-by-trial variations in momentary happiness during the task; the best-fitting model differentiated between partner reward prediction errors resulting from participant and partner choices, indicating that the impact of outcomes for the partner on participants' happiness varied depending on who chose. This confirms the importance of responsibility in determining the emotional consequences of risky social choices. fMRI analyses based on this computational model identified a left STS region responding more to partner reward prediction errors resulting from participant rather than partner choices. This suggests a critical role of the STS in monitoring the consequences of one's risky decisions on others, an essential social cognitive function. A last analysis showed that connectivity between this STS region and the right IFG varied depending on condition and choice, suggesting that the information processed in the STS is relayed to the IFG during social decisions. Our findings add to current understanding of the neural mechanisms underlying responsibility for, and guilt evoked by, the outcomes on others of social decisions under economic risk.

We used several approaches to compare choices made for self only or for both participant and partner. Participants made slightly more risk-seeking choices when deciding for themselves than for both themselves and the partner in Study 1, but this difference disappeared in Study 2. The $\rho$ parameter on which this finding in Study 1 is based could only be estimated in a minority of participants due to a relatively low number of trials, which suggests that this finding may not be very reliable. The simpler and more robust method (evaluation of a risk premium) showed no difference in risk aversion across conditions in either study. Overall, we believe that we do not have strong evidence of differences in risk preferences across conditions. Our findings are not unexpected given previous work. At least two studies reported that being responsible for somebody else's payoffs increases risk aversion (*Fareri et al., 2022*; *Pahlke et al., 2015*). However, participants in a recent neuroimaging study were more loss averse when choosing for themselves rather than known other people (*Arioli et al., 2023*). Indeed, recent large meta-analyses comparing risky choices for oneself vs. for others report either no difference in risk preferences (*Batteux et al., 2019*) or a small shift towards more risky decisions for others (*Polman and Wu, 2020*), with large variations across studies. Personal closeness to the others for which we decide seems to reduce differences in risk preferences (*Fareri et al., 2022*; *Zhang et al., 2017*), as does making decisions for self before making decisions for others (*Ifcher and Zarghamee, 2020*). In our study, decisions never only affected the partner, which most likely 'watered down' any differences in risk attitude in decisions for oneself vs. for someone else. We also created a friendly

prosocial environment in which people felt closer to each other than two strangers would (see Experiment Partner in Methods), and interleaved decisions for self only vs. for self and other. Similar risk preferences in decisions for self vs. self and other are thus unsurprising. The fact that participants made similarly risky decisions for themselves or for both during the fMRI study was to our advantage, because it made the neural signals evoked in these situations more comparable.

Anterior insula (aIns) response, particularly in the left hemisphere, was highest when partners received low lottery outcomes resulting from participants' risky choices. This replicates a large body of evidence associating aIns with feelings of guilt evoked during social decisions (see Introduction). Because we have neither asked our participants specifically what they felt in these situations, nor specifically whether they experienced guilt, we cannot exclude the possibility that they have instead or in addition felt empathy for their partner, a feeling of failure or bad luck, or some other emotion. The aIns region has also been associated with empathy for negative emotions such as disgust (*Wicker et al., 2003*) or pain (*Gu et al., 2012*; *Lamm et al., 2011*), with affective empathy during charitable giving (*Tusche et al., 2016*), and more generally with emotion awareness (*Bird et al., 2010*; *Gu et al., 2013*). These functions might be supported by interoception (*Craig, 2002*): posterior insula is thought to receive interoceptive information from the body and to pass it on to the anterior insula for integration with sensory, emotional, cognitive, and motivational signals from other regions (*Bid Craig, 2009*; *Rogers-Carter and Christianson, 2019*). This integration would allow one to represent one's own, as well as estimates of other people's, feelings and bodily states and allow error-based learning based on these (*Lamm and Singer, 2010*; *Rogers-Carter and Christianson, 2019*; *Singer et al., 2009*). Our finding of changing functional connectivity between aIns and prefrontal cortex as a function of decision taken and experimental condition during choice may reflect changing emotional information transfer between these structures depending on choice and social context. How we feel when we witness our decisions' consequences on others is an important signal to consider when attempting to make good social decisions. Unsurprisingly, lesions of aIns are associated with reduced altruistic attitudes (*Chau et al., 2018*), individuals with higher levels of psychopathic traits show reduced modulation of aIns response to anticipated guilt (*Seara-Cardoso et al., 2016*), and show reduced guilt aversion (*Gong et al., 2019*).

Deciding for both self and partner rather than just for oneself evoked increased activations in precuneus, left TPJ, and dmPFC, areas classically associated with social cognition (*Frith and Frith, 2006*; *Schurz et al., 2014*; *Van Overwalle, 2009*), and also with feelings of guilt (for meta-analyses, see *Gifuni et al., 2017*; *Piretti et al., 2023*). This replicates previous findings of engagement of dmPFC, STS, and/or TPJ when making decisions involving others (*Jung et al., 2013*; *Nicolle et al., 2012*; *Ogawa et al., 2018*; *Piva et al., 2019*). Interestingly, our computational model-based analysis revealed a cluster of voxels in the left STS that responded positively to partner prediction errors, but only when these resulted from decisions made by the participant instead of the partner. This is congruent with associations between STS and cognitive perspective taking during charitable giving (*Tusche et al., 2016*), representation of other people's interests during altruistic choice (*Hutcherson et al., 2015*), or more generally with mentalizing computations during strategic social choice (*Carter et al., 2012*; *Hampton et al., 2008*; *Hill et al., 2017*). The features of our experimental design (direct contrast of similar decisions made by the participant and partner; independent lottery outcomes for self and other; quantification of the consequences of decisions through variations of momentary happiness) allowed us to identify this very specific neural signal indicative of a neural sensitivity to the consequences of one's own actions on others, whether positive or negative. A neural signal coding partner reward prediction errors resulting from one's decisions seems essential to guide social decisions and as a basis for empathic concern for the people influenced by our actions. Anecdotally, we found a weak functional connection between this left STS cluster and left aIns that varied as a function of choice and experimental condition; this is interesting because connections between a region in the left TPJ and aIns have been shown to vary depending on people's tendency to seek or avoid responsibility for others (*Edelson et al., 2018*). Given the roles of the left STS/TPJ and aIns in guilt and social decisions, it is not surprising to find information exchanges between these structures. More experiments investigating specific computations in which these regions are involved (as in, e.g., *Charpentier and O'Doherty, 2018*; *Konovalov et al., 2021a*; *Konovalov and Ruff, 2021b*) are likely to help understand the neural mechanisms by which guilt and responsibility influence social decision-making.

There are several limitations to our study. The first limitation is that the partner's decisions were in fact taken by an algorithm. This was not communicated to the participants and was thus a case of deception, which is inadmissible in behavioural economics. To our defence, this study was planned as a social neuroscience study and was executed before we had taken into account the practices of behavioural economists. However, this approach did have the advantage of eliminating potentially complex iterative reciprocal influences of decisions and outcomes between the players, which would have led to much more complicated emotional states and decisions that would have been difficult to understand and model. Thus, from an analytical point of view, our approach might have actually allowed a cleaner comparison between decisions and outcomes than if the partner had really made the decisions. The fact that partner outcomes also influenced participants' momentary happiness demonstrates that participants were not emotionally detached from the consequences of their actions on their partner. This finding is, of course, essential for the validity of our results and suggests that the effects could get stronger with more direct interactions. Therefore, although we will abstain from using deceptive practices when pursuing this research, we believe our findings to be valid.

Another limitation is that we have not asked participants to specifically name emotions as they proceeded through the experiment. As a result, we cannot ascertain whether people experienced guilt, shame, regret, disappointment or another specific emotion during the experiment. However, asking about specific emotions is itself associated with several drawbacks: emotion labels might have coloured participants' spontaneous emotional state; participant responses might be influenced by social desirability; thinking about which emotion best characterizes one's mental state takes time and distracts from the decision task; and the list of emotions to choose from is necessarily limited. In any case, the precise naming of emotions was not the aim of our study; instead, we relied on a definition of guilt from psychological game theory (*Battigalli and Dufwenberg, 2007*). Indeed, previous work on social emotions recommends 'study the experiential phenomenology of emotions instead of mere emotion labels', because 'as a psychological explanation of human behaviour, the phenomenological experience of an emotion is much more important than the label attached to this experience' (*Zeelenberg and Breugelmans, 2008*). The 'neurometrics' analysis of our guilt-related activation maps showed a significant similarity to a published guilt-related brain signature (*Yu et al., 2020*). We thus believe that our participants did experience guilt when their choices led to low outcomes for their partner.

Several open questions remain at the end of this study. As discussed above, asking participants directly about which emotions they have felt during the different stages of this task would allow us to link subjective experience with our analytical measures. Testing more participants would allow us to assess the impact of inter-individual variations in personality traits on the experience as well as the behavioural and neural correlates of guilt and responsibility. Using more trials in the experiment would allow separate modelling of risk preferences in gain and loss trials in each experimental condition using expected utility models, and could allow testing whether changes in momentary happiness affect subsequent choices. Varying partner identities (friends, stranger, and artificial agent) could reveal the impact of social discounting on guilt and responsibility. Functional connectivity analyses could also be performed for the data obtained in the feedback part of the paradigm. In sum, we believe that this experimental approach lends itself very well to the study of several aspects of social emotions.

## Materials and methods

**Key resources table**

| Reagent type (species) or resource | Designation | Source or reference | Identifiers | Additional information |
|---|---|---|---|---|
| Software, algorithm | MATLAB | https://www.mathworks.com/products/matlab.html | RRID:SCR_001622 | Version R2016B and R2024A |
| Software, algorithm | Psychtoolbox | http://psychtoolbox.org | RRID:SCR_002881 | |
| Software, algorithm | SPM | http://psychtoolbox.org | RRID:SCR_007037 | SPM12 (7771) |
| Software, algorithm | R | https://www.r-project.org | RRID:SCR_001905 | Version 4.2.1 |

*Continued on next page*

*Continued*

| Reagent type (species) or resource | Designation | Source or reference | Identifiers | Additional information |
|---|---|---|---|---|
| Software, algorithm | lme4 | https://cran.r-project.org/package=lme4 | RRID:SCR_015654 | |
| Software, algorithm | JASP | https://jasp-stats.org | RRID:SCR_015823 | Version 0.16.1 |
| Software, algorithm | Measures of Effect Size Toolbox for Matlab | https://github.com/hhentschke/measures-of-effect-size-toolbox | RRID:SCR_014703 | |

## Participants

Forty healthy participants (14 male, mean age 26.1, range 22–31) participated in Study 1 (behaviour only study), and 44 healthy participants (19 male, mean (SD) age = 30.6 (6.5), range 23–50) participated in Study 2 (fMRI study). All participants provided written informed consent. Participants were recruited from the local population through advertisements on online blackboards at the University of Bonn and on local community websites, and through flyers posted in libraries, university cafeterias and sports facilities. The number of participants recruited in Study 2 corresponds to the sample size estimated using G*Power 3.1 software (*Faul et al., 2007*) for a two-way *t*-test assessing the difference between two means (matched pairs) based on the results of Study 1 (Cohen's *d* = 0.56), with alpha error = 0.05 and power (1 − beta) = 0.95, the required sample size was 44. The fMRI data from four participants in Study 2 were excluded from the fMRI data analysis because of excessive head motion (>3 mm or >3°).

## Experimental procedure

The design of the experiment was inspired by previous studies investigating how risky choices and their consequences influence momentary happiness (*Rutledge et al., 2014*; *Rutledge et al., 2016*). It was implemented in MATLAB (Version R2016b; RRID:SCR_001622; The MathWorks, Inc, Natick, MA) using the Psychtoolbox extensions (RRID:SCR_002881).

## Experiment partner

Participants played with a friendly same-sex experiment partner (pairs of participants in Study 1, authors TW and MG in Study 2), whom they met before the scan for an introduction session. In this session lasting about 15 min, participants played an ice breaker game with their experiment partner, in which both participant and partner took turns in drawing one half of a simple picture while being blindfolded and following verbal instructions given by their game partner. Encouragements and other positive feedback were given by the partner throughout the task. This icebreaker game led to an agreeable social atmosphere and positive, non-competitive, sympathetic attitude between the participants and the partner. Results of a brief questionnaire indicate that this approach was successful: participants' average ratings of their partners in terms of sympathy, cooperativity, honesty, openness and sociability were all above 8 on a scale of 1–10, in both studies (*Appendix 1—table 11*).

## Decision task

Following the icebreaker game, participants in Study 1 performed three sessions of a task in which they decided on each trial between a safe and a risky monetary option (*Figure 1*). The risky monetary option was a lottery with two equiprobable outcomes (lottery). There were three kinds of trials: decisions by the participant only for themselves (*Solo* condition), decisions by the participant for themselves and the partner (*Social* condition), and decisions by the partner for both themselves and the participant (*Partner* condition). Importantly, when the risky option was selected in the *Social* or *Partner* condition, the lottery was played out independently for the participant and the partner, such that both could receive the higher (*HighOutcome*) or lower outcome (*LowOutcome*), independently from each other. In order to ascertain constant decisions by the partner, the partner's decisions were simulated using a simple algorithm that always selected the option with the highest expected value; i.e., it selected the lottery if EV diff >0; *EVdiff = (high lottery outcome + low lottery outcome)/2 − safe outcome*. The amounts earned per trial were determined as in a previous study (*Rutledge et al., 2014*), as follows. There were 20 mixed trials, 20 gain trials, and 20 loss trials per session. In the mixed

trials, participants chose between a safe option of 0 € and a lottery consisting of a gain and a loss amount. Gain amounts were selected randomly out of the following: [15, 25, 40, 55, 75] cents. Loss options were determined by multiplying the gain amount with one randomly selected value out of the following: [-0.2, –0.34, −0.5, −0.64, −0.77, −0.89, −1, −1.1, −1.35, −2]. In gain trials, participants chose between a safe gain out of the following: [10, 15, 20, 25, 30] cents and a lottery with 0 € and a higher gain amount, calculated by multiplying the safe amount with one of the following values: [1.68, 1.82, 2, 2.22, 2.48, 2.8, 3.16, 3.6, 4.2, 5]. In loss trials, participants chose between a certain loss or a lottery with 0 € or a larger loss. Certain loss amounts were one of [–10, –15, −20,–25, −30] cents, and the larger loss amount of the lottery was a multiple of the certain loss amount and one of the following values: [1.68, 1.82, 2, 2.22, 2.48, 2.8, 3.16, 3.6, 4.2, 5]. The position of the safe and risky option on the screen (left or right) was determined randomly on every trial, and the different trial types (*Solo*, *Social*, or *Partner* condition; gain, loss and mixed trials) were presented in randomized order, with the constraint that there could not be more than two trials of the same condition in a row. Participants had unlimited time to choose between the safe or risky option. Decisions were displayed for 2 s and lottery outcomes for 2.5 s. Trials were separated in time by an inter-stimulus interval (ISI) of 1–2 s drawn randomly from a gamma distribution.

## Momentary happiness

Every two trials, one ISI after the outcome of the previous trial, participants were asked 'How happy are you right now?'. They could respond by selecting a value on a scale from 'very happy' (right) to 'very unhappy' (left) by moving a cursor with a button press. The start position of the cursor was the midpoint of the scale, and the scale had 100 selectable options. For analysis, happiness ratings were *Z*-scored to cancel out effects of different rating variabilities across participants.

## Difference between studies 1 (behaviour only) and 2 (fMRI study)

In Study 2, participants performed two sessions of the experiment described above inside the fMRI scanner. All parameters were identical except that ISIs varied from 3 to 11 s (drawn randomly from a gamma distribution). In Study 2, instead of playing against another same-sex participant, participants played either against experimenter MG or TW depending on sex (participant and partner were always of same sex) who were outside the scanner.

## Statistical analysis

### General information

Statistical analysis was performed using Matlab R2024A (RRID:SCR_001622) and the lme4 package (*Bates et al., 2022*) (RRID:SCR_015654) in R (version 4.2.1, RRID:SCR_001905), and JASP (version 0.16.1, RRID:SCR_015823, JASP Team 2021). fMRI data were analysed using SPM12 software (RRID:SCR_007037; Wellcome Trust Centre for Neuroimaging, London, UK). All statistical tests were two-tailed. Bayes factors were calculated using default priors in JASP and express the probability of the data given $H_1$ relative to $H_0$ ($BF_{10}$, values larger than one are in favour of H1). Effect sizes were calculated using standard approaches implemented in Matlab, including the Measures of Effect Size Toolbox (RRID:SCR_014703).

### Decisions

#### Generalized linear mixed logistic regression

First, we assessed whether choices between the safe and risky options were reasonable and whether these choices varied between *Solo* and *Social* conditions. To do so, we modelled choices with a mixed-effects logistic regression with subject-specific random intercepts $u_{0j} \sim N(0,\delta_u^2)$ and slopes $[u_{1j}, u_{2j}] \sim N(0,\delta_u^2)$ and residual $\varepsilon_{ij} \sim L(0,1)$, assumed to follow the standard logistic distribution. In this regression model, the choice of the risky option (Chose risky: 1 = Yes, 0 = No) was entered as the dependent variable, and explanatory variables were the difference between the expected value of the lottery minus the value of the safe option [*EVdiff* = (*high lottery outcome + low lottery outcome*)/2 − *safe outcome*] and condition (*Cond*: 1 = *Solo*, 2 = *Social*; a categorical variable), as well as the interaction between them (*EVdiff × Cond*). All explanatory variables were mean-centred. The random slopes are designed to account for between-subject heterogeneity:

$$\textit{Chose risky}_{ij} = 1 \textit{ if } \beta_0 + (\beta_1 + u_{1j})^* EVdiff_{ij} + (\beta_2 + u_{2j})^* Cond_{ij} + \beta_3{}^* EVdiff \times cond_{ij}$$
$$+ u_{0j} + \varepsilon_{ij} > 0, \text{ and } \textit{Chose safe}_{ij} = 0 \text{ otherwise.}$$

This gives rise to the regression equation:

$$\textit{Chose risky}_{ij} = F(\beta_0 + (\beta_1 + u_{1j})^* EVdiff_{ij} + (\beta_2 + u_{2j})^* Cond_{ij} + \beta_3{}^* EVdiff \times cond_{ij} + u_j), \qquad (1)$$

where $F(x) = 1/[1 + \exp(-x)]$ is the cumulative distribution function of the standard logistic distribution. The subscript $j$ indexes participants, while $i$ indexes observations per participant. An observation corresponds to one choice (*Chose risky*: 1 = Yes, 0 = No).

This model was estimated for Studies 1 and 2 separately, using the package *lme4* in R.

## Estimating risk attitude 1: risk premium

Next, we estimated risk attitudes by calculating a *risk premium* for each participant and condition, defined as the difference between the expected value of a lottery and its certainty equivalent (positive values indicate risk aversion, negative values risk seeking). The certainty equivalent is the smallest amount an individual would be indifferent to spending on a lottery. We estimated the certainty equivalent of the lotteries by fitting to each participant's decisions in each condition the logistic regression model with residual $\varepsilon_{ij} \sim L(0,1)$, assumed to follow the standard logistic distribution:

$$\textit{Chose risky}_i = F(\beta_0 + \beta_1{}^* EVdiff_i) \qquad (2)$$

The *risk premium* for each condition was defined as the EVdiff value corresponding to 50% *risky* choices indicated by the fitted logistic regression model. In both studies, these *risk premium* values were compared between *Solo* and *Social* conditions using paired *t*-tests.

## Estimating risk attitude 2: CARA model

Next, we estimated risk attitudes using expected utility theory. In this approach, an agent chooses between two options by comparing their expected utility values. Utility values reflect the agent's subjective values or preferences for a set of choice options, revealed by the agent's choices. Expected utility theory uses utility functions to describe the agents' preferences, including that agent's attitude towards risk. One simple model of risk aversion termed Constant Absolute Risk Aversion (CARA) assumes an exponential utility function:

$$U(x, \rho) = 1 - \mathrm{e}(-\rho x),$$

where $x$ is a monetary amount and $\rho$ is a constant that represents the degree of risk preference. $U$ has an upward slope for perceived gains, indicating higher utility for a larger amount received. For a risk-averse individual facing a choice in the domain of gains, $\rho$ is positive and $U$ is concave: a sure amount is preferred over a lottery with the same expected value. For a risk-seeking individual facing the same choices, $\rho$ is negative.

Under CARA, the expected utility of a lottery is:

$$EU_{lottery} = p^* U(x_1, \rho) + (1 - p)^* U(x_2, \rho)$$

where $x_1$ and $x_2$ are the two possible outcomes of the lottery and $p$ is the probability of obtaining outcome $x_1$.

The certainty equivalent of a lottery under CARA is:

$$CE_{lottery} = log(1 - \rho^* EU_{lottery})/ - \rho \qquad (3)$$

For $\rho \neq 0$.

Following an econometric method (***von Gaudecker et al., 2011***), we estimated $\rho$ by fitting a probit regression with a cumulative Gaussian link function $\Phi$ to choices obtained in the *Solo* and *Social* conditions of each participant:

$$\phi = 0.5^*(1 + erf((CE - VS)/(\sqrt{2}^* \sigma)));$$

where *CE* is the certainty equivalent of the lottery, *VS* the value of the safe alternative option, $\sigma$ is Fechner noise and erf is the error function. The regression was fitted using non-linear least-squares implemented in Matlab's nlinfit.m. In both studies, these $\rho$ values were compared between *Solo* and *Social* conditions using paired *t*-tests.

## Momentary happiness

### Computational models of happiness variations

Following previous work (***Rutledge et al., 2014***; ***Rutledge et al., 2016***), we modelled the variations in momentary well-being (happiness ratings) using computational models that considered the influence of recent reward expectations, prediction errors resulting from those expectations, and differences in rewards obtained by the game partners. All models contained separate terms for certain rewards, expected value for lotteries and reward prediction errors, with influences that decayed exponentially over trials. We ran five models: (1) a 'Basic' model (***Equation 4***; the winning model in ***Rutledge et al., 2014*** in which there was no *Social* condition) with certain rewards (*CR*), expected value of chosen lotteries (*EV*), participant's reward prediction errors (*sRPE*, where s stands for 'self') and a forgetting factor that makes events in more recent trials more influential than those in earlier trials (gamma); (2) an 'Inequality' model, in which one regressor modelled both disadvantageous and advantageous inequality (***Equation 5***); (3) a 'Guilt-envy' model, the winning model in the social task in ***Rutledge et al., 2016***, which contains separate additional regressors for disadvantageous and advantageous inequality (***Equation 6***); (4) a 'Responsibility' model in which in addition to the participant's reward prediction errors (*sRPE*) we also modelled the partner's reward prediction errors (*pRPE*, where p stands for partner), with a distinction between *pRPE*s resulting from participant choices (*social_pRPE*) and partner choices (*partner_pRPE*) (***Equation 7***); and (5) a 'Responsibility Redux' model identical to the precedent except for the absence of *partner_pRPE*s, as we realized that this regressor explained little variance (***Equation 8***). The Basic, Inequality and Guilt-envy models are identical to those used in ***Rutledge et al., 2016***.

The equations of these models were as follows.

The equation for the Basic model was:

$$Happiness\,(t) = w_1 \sum_{j=1}^{t} \gamma^{t-j} CR_j + w_2 \sum_{j=1}^{t} \gamma^{t-j} EV_j + w_3 \sum_{j=1}^{t} \gamma^{t-j} sRPE_j \tag{4}$$

where *t* is trial number, $\gamma$ is a forgetting factor ($0 \leq \gamma \leq 1$) that weighs events in more recent trials more heavily than events in earlier trials (exponential decay over time), and weights $w_1$ to $w_3$ capture the influence of the following different event types: $CR_j$ is the certain reward received when the safe option was chosen instead of the lottery on trial *j*, $EV_j$ is the average reward for the lottery if chosen on trial *j* and $sRPE_j$ is the participant's reward prediction error on trial *j* obtained as a result of choosing the lottery. Terms for unchosen options were set to zero. No constant term was used as ratings were Z-scored, resulting in a mean of the data of 0.

As described above, we complemented the Basic model by adding terms representing additional influences on variations in participant happiness. Two models included regressors accounting for inequality, and two models included regressors accounting for partner reward prediction errors. The equation for the first of the inequality models, the 'Inequality' model, was as follows:

$$\begin{aligned} Happiness\,(t) \quad &= w_1 \sum_{j=1}^{t} \gamma^{t-j} CR_j + w_2 \sum_{j=1}^{t} \gamma^{t-j} EV_j + w_3 \sum_{j=1}^{t} \gamma^{t-j} sRPE_j \\ &+ w_4 \sum_{j=1}^{t} \gamma^{t-j} \max\left(|S_j - P_j|, 0\right)_j \end{aligned} \tag{5}$$

Here, $S_j$ and $P_j$ are the rewards received by the participant and by their experiment partner, respectively, and thus $w_4$ relates to the influence of inequality of any kind on the variation in happiness. The equation of the 'Guilt-envy' model was as follows:

$$Happiness\,(t) \quad = w_1 \sum_{j=1}^{t} \gamma^{t-j} CR_j + w_2 \sum_{j=1}^{t} \gamma^{t-j} EV_j + w_3 \sum_{j=1}^{t} \gamma^{t-j} sRPE_j$$
$$+ w_4 \sum_{j=1}^{t} \gamma^{t-j} \max\left(S_j - P_j, 0\right)_j + w_5 \sum_{j=1}^{t} \gamma^{t-j} \max\left(P_j - S_j, 0\right)_j \tag{6}$$

Here, $w_4$ relates to advantageous inequality and $w_5$ relates to disadvantageous inequality, modelled separately. The next two models included terms relating to the partner's RPE.

The equation for the 'Responsibility' model was:

$$Happiness\,(t) \quad = w_1 \sum_{j=1}^{t} \gamma^{t-j} CR_j + w_2 \sum_{j=1}^{t} \gamma^{t-j} EV_j + w_3 \sum_{j=1}^{t} \gamma^{t-j} sRPE_j$$
$$+ w_4 \sum_{j=1}^{t} \gamma^{t-j} social\_pRPE_j + w_5 \sum_{j=1}^{t} \gamma^{t-j} partner\_pRPE_j \tag{7}$$

In this model, the influence of the partner's RPE resulting from participant choices in the *Social* condition (*social_pRPE*) is modelled separately from the influence of the partner's RPE resulting from partner choices, in the *Partner* condition (*partner_pRPE*). The equation for the last model, 'Responsibility Redux', was the same as for the Responsibility model (*Equation 7*), except that we omitted the regressor coding partner's RPE resulting from partner choices, as they had a smaller impact on participant happiness. This led to the following equation:

$$Happiness\,(t) \quad = w_1 \sum_{j=1}^{t} \gamma^{t-j} CR_j + w_2 \sum_{j=1}^{t} \gamma^{t-j} EV_j + w_3 \sum_{j=1}^{t} \gamma^{t-j} sRPE_j$$
$$+ w_4 \sum_{j=1}^{t} \gamma^{t-j} social\_pRPE_j \tag{8}$$

## Likelihood ratio test

To formally assess which of our models fitted the data best, we supplemented the AIC, BIC, $R^2$ and adjusted $R^2$ values reported in *Table 1* with a series of *likelihood ratio tests*: we compared pair-wise the likelihoods of the *Responsibility* model given the data to the likelihoods of all the other models. The equation for this test was the following:

$$LR = 2{}^{*}\log(L_0/L_1);$$

where $L_0$ and $L_1$ are, respectively, the likelihoods of the simpler and more complex model. The likelihood ratio follows a Chi-square distribution, with degrees of freedom equal to the number of extra parameters in the more complex model. The equation simplifies to:

$$LR = -2{}^{*}(\log(L_1) - \log(L_0)) \tag{9}$$

## Parameter recovery

Stability of the estimated parameters of these models was evaluated by attempting to recover parameters from synthetic data created using each participant's real estimated parameters. After fitting each participant's momentary happiness data (see above), we synthesized new momentary happiness data based on each participant's estimated parameters, added 1SD of noise to the happiness data, fitted the model to these synthetic data, and repeated this procedure 10 times, for both studies. We then compared these new estimated parameters to the actual parameters from which the synthetic data were generated, as follows: For each parameter, we calculated the mean of each participant's recovered parameters and regressed these means on the participants' actual parameters (see *Figure 3— figure supplement 1*). The results show that the estimated parameters could be reliably recovered from noisy synthetic data.

## LMM regressions of happiness variations

To better understand the impact of responsibility for outcomes of risky social choices on happiness, we focused on the happiness reported after outcomes of lottery choices in *Social* and *Partner* trials and evaluated the impact of who chose the lottery. We performed this analysis using a mixed-effects linear regression analysis with residual $\varepsilon_{ij} \sim N(0, \sigma\varepsilon^2)$ and subject-specific random intercepts $uj \sim N(0,$

$\sigma u^2$). Specifically, we regressed Z-scored momentary happiness ratings reported after these outcomes on dummy variables representing occurrences (coded as 1 if event present, 0 otherwise) of the participant receiving the high lottery outcome, the partner receiving the high lottery outcome, the participant having chosen the lottery, and interactions between these terms.

$$
\begin{aligned}
\text{Happiness(t)} \quad &= 1 + \beta_1 sHigh_{ij} + \beta_2 pHigh_{ij} + \beta_3 sDec_{ij} + \beta_4 sHigh^* pHigh_{ij} + \beta_5 sHigh^* sDec_{ij} \\
&\quad + \beta_6 pHigh^* sDec_{ij} + uj + \varepsilon ij.
\end{aligned}
\tag{10}
$$

sHigh and pHigh had a value of 1 if the participant or the partner (respectively) received the high lottery outcome and 0 otherwise, sDec had a value of 1 if the participant chose the lottery. All models were estimated for Studies 1 and 2 separately, using the *lme4 package in R*. In both studies, the model reported in *Equation 10* fitted the data significantly better ($p < 2e-5$) than the simpler models without interactions (higher total and adjusted $R^2$, see *Appendix 1—table 2*), but not significantly worse than the model with all interactions ($p > 0.5$; tested with the ANOVA function in R). The regressors of the different models tested and their relative effects are reported in *Appendix 1—table 2*.

## fMRI data acquisition and pre-processing

Imaging data were collected on a 3T Siemens TRIO MRI system (Siemens AG, Erlangen, Germany) with a Siemens 32-channel head coil. Functional data were acquired using a T2* echo-planar imaging (EPI) BOLD sequence, with a repetition time (TR) of 2500 ms, an echo time (TE) of 30 ms, 37 slices with voxel sizes of 2 × 2 × 3 mm, a flip angle of 90°, a field of view of 192 mm and PAT two acceleration. To exclude participants with apparent brain pathologies and facilitate normalization of the functional data, a high-resolution T1-weighted structural image was acquired, with a TR of 1660 ms, a TE of 2540 ms, 208 slices with voxel sizes of 0.8 × 0.8 × 0.8 mm and a field of view of 256 mm. Data were then pre-processed and analysed as in previous studies (e.g., *Schultz et al., 2019*) using standard procedures in SPM12. The first five volumes of each functional time series were discarded to allow for T1 signal equilibration. The structural image of each participant was coregistered with the mean functional image of that participant. Functional images were corrected for head movement between scans by a 6-parameter affine realignment to the first image of the time-series and then re-realigned to the mean of all images. The structural scan of each participant was spatially normalized to the current Montreal Neurological Institute template (MNI305) by segmentation and non-linear warping to reference tissue probability maps in MNI space, and the resulting normalization parameters were applied to all functional images which were then resampled at 2 × 2 × 2 mm voxel size, then smoothed using an 8 mm full width at half maximum Gaussian kernel. Time series were de-trended by the application of a high-pass filter (cut-off period, 128 s).

## fMRI data analysis

The pre-processed functional data were analysed using a standard two-stage approach based on the general linear model (GLM) implemented in SPM12 running in MATLAB, with corrections for multiple comparisons. Individual participants' data were modelled with fixed effects models, and their summary data were entered in random effects models for group statistics and inferences at the population level.

### Fixed-effects models

The fixed effects models implemented a mass univariate analysis applied to the pre-processed data. The first GLM (GLM1) included the following event types for each session: presentation of choice options, safe choice, risky choice (=lottery), receipt of outcome of safe option, all modelled separately for the *Social*, *Partner*, and *Solo* conditions; then receipt of high lottery outcome by the partner, separately coded for the *Social* and *Partner* conditions, and receipt of high lottery outcome by the participant; then receipt of low lottery outcome by the partner, separately coded for the *Social* and *Partner* conditions, and receipt of low lottery outcome by the participant; a final regressor modelled the button presses that occurred when reporting momentary happiness. All events were modelled as stick functions at the time of occurrence and convolved by SPM using the canonical haemodynamic function. In addition, we created another GLM (GLM2) with regressors designed to identify brain regions whose activation reflected the variables of the best-fitting

computational model (see above). This model contained regressors coding the receipt of certain rewards (*CR*), the expected value of the chosen option (*EV*), the participant's reward prediction error (*sRPE*), the partner RPE resulting from decisions made by the participant (*social_pRPE*) and the partner RPE resulting from decisions made by the partner (*partner_pRPE*). Those regressors were applied as follows to each experimental session of each participant: (1) fitting the computational model to the momentary happiness data; (2) creation of a series of stick functions representing the time of occurrence of the modelled events during the scan, with stick magnitude determined by the fitted value calculated by the model; and (3) convolution of the series of stick functions with the canonical BOLD function (as implemented in SPM). The 5 computational regressors were entered as covariates into each participant's model, which also contained one regressor modelling the button presses that occurred when reporting momentary happiness. All GLMs contained six motion-correction parameters included as regressors of no-interest to account for motion-related artefacts. Regression coefficients (parameter estimates) were estimated for each regressor in each voxel of each participant's brain.

## PPI analysis
We used the gPPI toolbox (**McLaren et al., 2012**) for SPM to run seed-to-voxel functional connectivity analyses. We ran two models for each participant, both based on GLM1, with the following seeds: the left insula cluster more sensitive to the outcomes of *Risky* vs. *Safe* choices, yielding GLM3 (see **Figure 4D, E**), and the left STS cluster responding more to *social_pRPE* than *partner_pRPE*, yielding GLM4 (**Figure 4H**). Seeds were identical for all participants. PPI models thus contained the same regressors as GLM1, plus the interaction between the first eigenvariate of the signal in the seed region and the regressors coding the experimental conditions of interest: *Safe* and *Risky* choice in *Social*, *Partner* and *Solo* conditions.

## Random-effects models
Individual parameter estimate maps of the responses to the experimental conditions were assessed in random-effects tests to evaluate effects at the level of the group of participants. These tests were second-level models based on individual parameter estimate maps. We tested six models: a full factorial model to investigate the response during decisions [factors *Choice* (*Safe* or *Risky*) and *Condition* (*Social* or *Solo*), based on GLM1 estimate maps]; a full factorial model to investigate the response during the receipt of outcomes [factors *Outcome* (*Safe*, *HighOutcome*, or *LowOutcome*) and *Condition* (*Social*, *Partner*, or *Solo*), based on GLM1 estimate maps]; a paired *t*-test contrasting negative lottery outcomes for the partner resulting from participant or partner choices (based on GLM1 estimate maps); a model to investigate the response to the computational regressors (*CR*, *EV*, *sRPE*, *social_pRPE*, and *social_pRPE*, based on GLM2 estimate maps); and two full factorial models to investigate functional connectivity during decisions [factors *Choice* (*Safe* or *Risky*) and *Condition* (*Social* or *Solo*), based on estimate maps of GLM3 and GLM4].

## Linear mixed models
Similar to the behavioural data, we analysed GLM1 parameter estimates from voxels of specific regions of interest (see Results) using mixed-effects linear regressions with residual $\varepsilon_{ij} \sim N(0, \sigma\varepsilon^2)$ and subject-specific random effects $uj \sim N(0, \sigma u^2)$. These models were implemented in Matlab and fitted with the fitlme.m function using restricted maximum likelihood estimation. We regressed parameter estimates on dummy variables representing variables of interest. For responses during the decision phase of the experiment, our models included factors *Choice* (*Safe* or *Risky*), *Condition* (*Social*, *Partner*, or *Solo*), *Run* (run one or two), and *ROI*, with all interactions. For responses during the outcome phase, our models included the factors *Social* (value of 1 for the *Social* condition and 0 for the *Partner* condition), *LowOutcome* (value of 1 for *Low* lottery outcome and 0 for *High* lottery outcome), *Run*, with all interactions. Models without the factor *Run* were tested as well, and the best-fitting models were tested further. In all models, *Subject* was the only random factor (random intercept). To better understand significant interaction effects, we ran subsequent analyses on contrasts of parameter estimates (see Results and Appendix 1).

## Thresholding

All clusters reported survive a significance threshold of p < 0.05 with family-wise error correction for multiple comparisons across the whole brain or a smaller volume when explicitly mentioned, based on an uncorrected voxel-wise (cluster-forming) threshold of $p < 0.001$.

## Acknowledgements

We would like to thank R Rutledge for discussions and for providing the reinforcement learning code used in their previous publication (*Rutledge et al., 2016*), H Gerhardt for advice and help using linear mixed models, logistic regressions and economic models, as well as M Heinrichs, T Kalenscher, and C Ruff for discussions. This work was supported by the Open Access Publication Fund of the University of Bonn.

## Additional information

### Funding

No external funding was received for this work.

### Author contributions

Maria Gädeke, Data curation, Software, Investigation, Methodology, Writing - original draft, Writing - review and editing; Tom Eric Willems, Data curation, Software, Investigation, Methodology; Omar Salah Ahmed, Software, Methodology; Bernd Weber, Resources, Funding acquisition; Rene Hurlemann, Conceptualization, Resources, Funding acquisition; Johannes Schultz, Conceptualization, Data curation, Software, Formal analysis, Supervision, Visualization, Methodology, Writing - original draft, Project administration, Writing - review and editing

### Author ORCIDs

Tom Eric Willems ⓘ https://orcid.org/0000-0001-9872-0724
Johannes Schultz ⓘ https://orcid.org/0000-0003-4117-232X

### Ethics

The studies fulfilled all relevant ethical regulations and were approved by the local ethics committee of the Medical Faculty of the University of Bonn, Germany (approval number 098/18). All participants gave written informed consent and the studies were conducted in accordance with the latest revision of the Declaration of Helsinki. Participants were remunerated for their time (10 Euros/hr) and received game earnings (0–10 Euros).

Reviewer #1 (Public review): https://doi.org/10.7554/eLife.105391.3.sa1
Reviewer #2 (Public review): https://doi.org/10.7554/eLife.105391.3.sa2
Author response https://doi.org/10.7554/eLife.105391.3.sa3

## Additional files

### Supplementary files
MDAR checklist

### Data availability

All behavioural data and group fMRI results maps needed to reproduce all the results and the figures of this study are freely available at GitHub (copy archived at *Schultz, 2026*). The entire raw dataset (anonymous data) is publicly available on OpenNeuro. All the code used to replicate the experiment, analyse the data and produce the figures of this study is freely available at GitHub (*Schultz, 2026*).

The following previously published dataset was used:

| Author(s) | Year | Dataset title | Dataset URL | Database and Identifier |
|---|---|---|---|---|
| Schultz J | 2024 | SoDec - Responsibility fMRI experiment dataset | https://doi.org/10.18112/openneuro.ds005588.v1.0.0 | OpenNeuro, 10.18112/openneuro.ds005588.v1.0.0 |

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

## Appendix 1

**Appendix 1—table 1.** Mixed-effects regressions on choices.
For details of the models, see Results and Methods in the main text. ***p < 0.001; **p < 0.01; *p < 0.05.

|  | Study 1 probit | Study 1 linear | Study 2 probit | Study 2 linear |
|---|---|---|---|---|
|  | 0.10 | 0.53*** | –0.04 | 0.49*** |
| (Intercept) | (0.08) | (0.02) | (0.07) | (0.02) |
|  | 0.07*** | 0.02*** | 0.09*** | 0.02*** |
| $EV_{risky} - V_{safe}$ | (0.01) | (0.00) | (0.01) | (0.00) |
|  | 0.14* | 0.03^ | 0.01 | 0.01 |
| Condition Social | (0.06) | (0.02) | (0.06) | (0.02) |
|  | –0.00 | –0.00 | –0.00 | 0.00 |
| $(EV_{risky} - V_{safe})$ * Condition | (0.00) | (0.00) | (0.01) | (0.00) |
| $R^2$ (ord) | 1.000 | 0.284 | 1.000 | 0.306 |
| $R^2$ (adj) | 1.000 | 0.283 | 1.000 | 0.305 |
| AIC | 4909 | 5395 | 3864 | 4313 |
| BIC | 4974 | 5466 | 3927 | 4382 |
| LogLikelihood | –2445 | –2687 | –1922 | –2146 |
| N | 4680 | 4680 | 3833 | 3833 |

**Appendix 1—table 2.** Linear mixed model regressions on happiness data following lottery choices. We fitted several models to the data in order to assess the stability of the effects. In both studies, Model 5 (*Equation 9* in the Results section of the main text), which contained all three two-way interaction terms, explained the data best, so its parameters for the crucial partnerHigh:participantDecided interaction are reported in the main text. All models contained the three main fixed effects and subject as a random effect (random intercepts), Models 2–4 contained one or two interaction terms. ***p < 0.001; **p < 0.01; *p < 0.05.

**Study 1**

|  | Model 1 | Model 2 | Model 3 | Model 4 | Model 5 |
|---|---|---|---|---|---|
| (Intercept) | –0.58*** | –0.57*** | –0.47*** | –0.47*** | –0.38*** |
|  | (0.06) | (0.06) | (0.06) | (0.07) | (0.07) |
| participantHigh [0,1] | 0.96*** | 0.95*** | 0.95*** | 0.94*** | 0.76*** |
|  | (0.05) | (0.07) | (0.05) | (0.07) | (0.09) |
| partnerHigh [0,1] | 0.43*** | 0.43*** | 0.23** | 0.23** | 0.05 |
|  | (0.05) | (0.05) | (0.07) | (0.07) | (0.09) |
| participantDecided [0,1] | –0.17** | –0.18* | –0.37*** | –0.39*** | –0.38*** |
|  | (0.05) | (0.08) | (0.08) | (0.09) | (0.09) |
| participantHigh:participantDecided | - | 0.02 | - | 0.02 | 0.02 |
|  |  | (0.11) |  | (0.11) | (0.1) |
| partnerHigh:participantDecided | - | - | 0.40*** | 0.40*** | 0.39*** |
|  |  |  | (0.11) | (0.11) | (0.1) |

*Appendix 1—table 2 Continued*

**Study 1**

| | | | | | |
|---|---|---|---|---|---|
| participantHigh:partnerHigh | - | - | - | - | 0.35*** |
| | | | | | (0.1) |
| *R*² (ord) | 0.25 | 0.25 | 0.259 | 0.259 | 0.266 |
| *R*² (adj) | 0.248 | 0.247 | 0.256 | 0.256 | 0.262 |
| *N* | 1216 | 1216 | 1216 | 1216 | 1216 |

**Study 2**

| | Model 1 | Model 2 | Model 3 | Model 4 | Model 5 |
|---|---|---|---|---|---|
| (Intercept) | –0.54*** | –0.50*** | –0.45*** | –0.43*** | –0.26** |
| | (0.06) | (0.07) | (0.07) | (0.08) | (0.09) |
| participantHigh [0,1] | 1.01*** | 0.95*** | 1.02*** | 0.96*** | 0.65*** |
| | (0.06) | (0.09) | (0.06) | (0.09) | (0.11) |
| partnerHigh [0,1] | 0.35*** | 0.35*** | 0.20* | 0.20* | –0.11 |
| | (0.06) | (0.06) | (0.09) | (0.09) | (0.11) |
| participantDecided [0,1] | –0.22*** | –0.28** | –0.38*** | –0.43*** | –0.45*** |
| | (0.06) | (0.09) | (0.09) | (0.11) | (0.11) |
| participantHigh:participantDec. | - | 0.11 | - | 0.1 | 0.12 |
| | | (0.12) | | (0.12) | (0.12) |
| partnerHigh:participantDecided | - | - | 0.29* | 0.28* | 0.31** |
| | | | (0.12) | (0.12) | (0.12) |
| participantHigh:partnerHigh | - | - | - | - | 0.58*** |
| | | | | | (0.12) |
| *R*² (ord) | 0.28 | 0.281 | 0.285 | 0.285 | 0.304 |
| *R*² (adj) | 0.278 | 0.278 | 0.282 | 0.281 | 0.299 |
| *N* | 876 | 876 | 876 | 876 | 876 |

**Appendix 1—table 3.** Details of clusters with activation varying as a function of choice and condition, during the decision phase (output of second-level SPM model plus Cohen's *d* for each cluster).

| Anatomy | Size (*N* vox.) | *T* | *d* | *Z* | MNI | | |
|---|---|---|---|---|---|---|---|
| | | | | | *x* | *y* | *z* |
| **Risky > safe** | | | | | | | |
| VStriatum R | 385 | 7.14 | 0.79 | 6.78 | 10 | 12 | –4 |
| VStriatum R | - | 5.28 | - | 5.12 | 14 | 22 | 4 |
| VStriatum L | 393 | 6.15 | 0.62 | 5.92 | –10 | 8 | –6 |
| **Social > solo** | | | | | | | |
| Precuneus R | 737 | 5.72 | 0.63 | 5.53 | 0 | –62 | 38 |
| Precuneus L | - | 5.17 | - | 5.02 | –10 | –56 | 34 |
| Precuneus L | - | 4.91 | - | 4.79 | –2 | –54 | 34 |
| Angular L (TPJ) | 320 | 4.50 | 0.55 | 4.40 | –34 | –58 | 26 |
| Temporal Sup L | - | 3.59 | - | 3.54 | –52 | –56 | 20 |

*Appendix 1—table 3 Continued on next page*

*Appendix 1—table 3 Continued*

| Anatomy | Size (*N* vox.) | *T* | *d* | *Z* | | MNI | |
|---|---|---|---|---|---|---|---|
| Medial PFC R | 302 | 3.97 | 0.54 | 3.90 | 4 | 52 | 22 |
| Medial PFC L | - | 3.97 | - | 3.90 | −6 | 58 | 22 |
| Medial PFC L | - | 3.63 | - | 3.57 | −6 | 54 | 32 |

**Appendix 1—table 4.** Linear mixed model regressions on BOLD response parameter estimates obtained during the decision phase – response during choice.

The parameter estimates of all voxels of the ROIs identified using the contrasts Risky > Safe and Social > Solo (see above and main text) were fitted with linear mixed models. The parameters of the best-fitting model (lowest BIC) for each ROI are reported below. We note here that the *Run* factor and interactions with it had significant effects in several ROIs, which shows that some of the effects reported varied across runs. However, for the sake of brevity, we will not discuss these results further. \*\*\*p < 0.001; \*\*p < 0.01; \*p < 0.05.

| | VStria L | VStria R | MPFC | Precun | TPJ L |
|---|---|---|---|---|---|
| (Intercept) | −0.78\*\*\* | −1.52\*\*\* | 1.18\*\*\* | 1.39\*\*\* | 0.94\*\*\* |
| | (0.13) | (0.13) | (0.18) | (0.18) | (0.14) |
| choice | −0.26\*\*\* | 0.78\*\*\* | −0.38\*\*\* | 0.97\*\*\* | 0.55\*\*\* |
| | (0.08) | (0.07) | (0.09) | (0.07) | (0.08) |
| condition | 0.01 | 0.34\*\*\* | −0.63\*\*\* | −0.40\*\*\* | −0.26\*\*\* |
| | (0.03) | (0.02) | (0.03) | (0.02) | (0.03) |
| run | 0.32\*\*\* | 0.96\*\*\* | −0.26\*\*\* | 0.30\*\*\* | 0.09\* |
| | (0.04) | (0.03) | (0.04) | (0.03) | (0.04) |
| choice:condition | 0.47\*\*\* | 0.04 | 0.12\*\* | −0.30\*\*\* | −0.24\*\*\* |
| | (0.04) | (0.03) | (0.04) | (0.03) | (0.04) |
| choice:run | 0.33\*\*\* | −0.29\*\*\* | 0.09 | −0.56\*\*\* | −0.37\*\*\* |
| | (0.05) | (0.05) | (0.06) | (0.05) | (0.05) |
| condition:run | −0.01 | −0.22\*\*\* | 0.17\*\*\* | −0.02 | −0.02 |
| | (0.02) | (0.02) | (0.02) | (0.01) | (0.02) |
| choice:condition:run | −0.18\*\*\* | 0.05\* | −0.01 | 0.16\*\*\* | 0.12\*\*\* |
| | (0.02) | (0.02) | (0.03) | (0.02) | (0.02) |
| $R^2$ (ord) | 0.136 | 0.158 | 0.208 | 0.170 | 0.175 |
| $R^2$ (adj) | 0.136 | 0.158 | 0.208 | 0.170 | 0.175 |

**Appendix 1—table 5.** Linear mixed model regressions on BOLD response parameter estimates obtained during the decision phase – difference between Risky and Safe choices, Social vs. Solo.

To better understand the choice:condition interaction, which was significant in all ROIs except the right striatum, we subtracted the response to safe choices from the response to risky choices for the four remaining ROIs and submitted these differences to additional linear mixed models, as above. The first model contained a factor socialVsSolo, in which data from the social condition were weighted positively, and trials in the solo condition were weighted negatively. As above, we tested these models both with and without the factor *Run* and associated interaction, and we report the best-fitting model in the table below: a dash ('-') in the row displaying parameters for the *run* and *socialVsSolo:run* regressors indicates that the model without factor *run* was better-fitting for this ROI.

| | VStria L | MPFC | Precun | TPJ L |
|---|---|---|---|---|
| (Intercept) | 0.67*** | –0.03 | 0.38*** | 0.07 |
| | (0.11) | (0.10) | (0.11) | (0.09) |
| socialVsSolo | –0.47*** | –0.10*** | 0.30*** | 0.24*** |
| | (0.03) | (0.01) | (0.02) | (0.03) |
| run | –0.03* | - | –0.24*** | –0.14*** |
| | (0.02) | | (0.01) | (0.01) |
| socialVsSolo:run | 0.18*** | - | –0.16*** | –0.12*** |
| | (0.02) | | (0.01) | (0.02) |
| $R^2$ (ord) | 0.085 | 0.075 | 0.068 | 0.091 |
| $R^2$ (adj) | 0.085 | 0.075 | 0.067 | 0.091 |
| N | 94,320 | 72,480 | 176,880 | 76,800 |

**Appendix 1—table 6.** Linear mixed model regressions on BOLD response parameter estimates obtained during the decision phase – difference between Risky and Safe choices, Social vs. Partner. Finally, we repeated this analysis with models containing a factor socialVsPartner, in which data from the social condition were weighted positively, and trials in the partner condition were weighted negatively. Here again, we report the best-fitting model from the versions with and without the factor *run*.

| | VStria L | MPFC | Precun | TPJ L |
|---|---|---|---|---|
| (Intercept) | 0.61*** | –0.03 | 0.38*** | 0.07 |
| | (0.11) | (0.10) | (0.11) | (0.09) |
| socialVsPartner | –0.07*** | 0.23*** | 0.39*** | 0.07** |
| | (0.01) | (0.01) | (0.02) | (0.03) |
| run | - | - | –0.24*** | –0.14*** |
| | | | (0.01) | (0.01) |
| socialVsPartner:run | - | - | –0.20*** | 0.01 |
| | | | (0.01) | (0.02) |
| $R^2$ (ord) | 0.090 | 0.225 | 0.233 | 0.228 |
| $R^2$ (adj) | 0.090 | 0.225 | 0.233 | 0.228 |
| N | 94,320 | 72,480 | 176,880 | 76,800 |

**Appendix 1—table 7.** Details of clusters with higher activation during risky vs. safe outcomes (second-level SPM model, with Cohen's *d* for each cluster).
Note: A dash ('-') in the *Size* or *d* column indicates that the peak reported on that line is part of a cluster whose centre is the next peak without dash listed above it.

| Anatomy | Size (*N* v.) | T | d | Z | x | y | z |
|---|---|---|---|---|---|---|---|
| Insula R | 1586 | 11.87 | 2.27 | Inf | 30 | 22 | –10 |
| | - | 10.51 | - | Inf | 42 | 22 | –8 |
| | - | 7.11 | - | 6.99 | 50 | 22 | 6 |
| Insula L | 1164 | 11.8 | 1.98 | Inf | –30 | 20 | –10 |
| | - | 6.15 | - | 6.07 | –50 | 16 | 8 |

*Appendix 1—table 7 Continued on next page*

*Appendix 1—table 7 Continued*

| Anatomy | Size (*N* v.) | *T* | *d* | *Z* | *x* | *y* | *z* |
|---|---|---|---|---|---|---|---|
| Dorsomedial_PFC | 2962 | 9.70 | 2.38 | Inf | 2 | 42 | 36 |
| | - | 8.31 | - | Inf | 2 | 20 | 60 |
| | - | 7.99 | - | 7.81 | 4 | 40 | 20 |
| Temp_Mid_R (STS) | 80 | 6.06 | 1.14 | 5.98 | 48 | −24 | −8 |
| | - | 4.93 | - | 4.89 | 50 | −34 | −2 |
| VStria_L | 30 | 5.73 | 1.23 | 5.66 | −10 | 0 | −6 |
| VStria_R | 40 | 5.68 | 1.18 | 5.62 | 8 | 4 | 2 |
| Parietal_Inf_R | 193 | 5.64 | 1.55 | 5.58 | 40 | −48 | 46 |
| | - | 5.12 | - | 5.07 | 36 | −60 | 54 |
| | - | 5.09 | - | 5.05 | 48 | −36 | 48 |
| DSL_PFC_R | 56 | 5.51 | 1.38 | 5.46 | 42 | 38 | 24 |
| Parietal_Inf_L | 64 | 5.27 | 1.25 | 5.21 | −46 | −44 | 46 |
| | - | 5.03 | - | 4.99 | −38 | −42 | 38 |

**Appendix 1—table 8.** Linear mixed model regressions on BOLD response parameter estimates obtained during the outcome phase – *Risky* minus *Safe* outcomes.

The parameter estimates of all voxels of the nine ROIs identified using the contrasts Risky > Safe outcome (see main Text) were fitted with linear mixed models. The parameters of the best-fitting model (lowest BIC) for each ROI are reported below. *Social* was a dummy variable with the value of 1 for the *Social* condition and 0 for the *Partner* condition; *LowOutcome* was a dummy with the value of 1 for *Low lottery outcome* and 0 for *High lottery outcome; Run* was a dummy with the value of 1 for run 1 and 2 for run 2. Subject was the only random factor. We note here that the *Run* factor and interactions with it were significant in several ROIs, which indicates that some of the effects reported varied across runs. However, for the sake of brevity, we will not discuss these results further. ***p < 0.001; **p < 0.01; *p < 0.05; ^p < 0.1.

| | VStriaL | VStriaR | InsulaL | InsulaR | STS_R |
|---|---|---|---|---|---|
| (Intercept) | 1.79*** | 1.12*** | 0.73*** | 0.52*** | 1.09*** |
| | (0.25) | (0.27) | (0.14) | (0.15) | (0.18) |
| Social | −1.30*** | −0.68*** | 0.42*** | 0.42*** | 0.21^ |
| | (0.22) | (0.08) | (0.04) | (0.03) | (0.12) |
| LowOutcome | −1.14*** | −0.12 | 0.83*** | 0.58*** | 0.51*** |
| | (0.22) | (0.08) | (0.04) | (0.03) | (0.12) |
| Run | −0.82*** | - | 0.18*** | 0.34*** | 0.40*** |
| | (0.10) | | (0.02) | (0.02) | (0.06) |
| Social:LowOutcome | 0.95** | 0.77*** | −0.76*** | −0.25*** | −0.81*** |
| | (0.31) | (0.12) | (0.06) | (0.05) | (0.18) |
| Social:Run | 0.89*** | - | −0.30*** | −0.29*** | −0.67*** |
| | (0.14) | | (0.03) | (0.02) | (0.08) |
| LowOutcome:Run | 0.60*** | - | −0.75*** | −0.43*** | −0.66*** |
| | (0.14) | | (0.03) | (0.02) | (0.08) |
| Social:LowOutcome:Run | −0.30 | - | 0.80*** | 0.30*** | 0.99*** |
| | (0.20) | | (0.04) | (0.03) | (0.11) |

*Appendix 1—table 8 Continued on next page*

*Appendix 1—table 8 Continued*

|  | VStriaL | VStriaR | InsulaL | InsulaR | STS_R |
|---|---|---|---|---|---|
| $R^2$ (ord) | 0.200 | 0.201 | 0.086 | 0.099 | 0.170 |
| $R^2$ (adj) | 0.200 | 0.201 | 0.086 | 0.099 | 0.170 |
| N | 9600 | 12,800 | 372,480 | 507,520 | 25,600 |

|  | ParL | ParR | dmPFC | FrontR |
|---|---|---|---|---|
| (Intercept) | –0.22 | 0.96*** | 0.73*** | –0.22 |
|  | (0.21) | (0.22) | (0.10) | (0.21) |
| Social | 0.14 | –0.60*** | –0.48*** | –0.07 |
|  | (0.13) | (0.09) | (0.02) | (0.15) |
| LowOutcome | 0.74*** | –0.81*** | 0.16*** | –0.80*** |
|  | (0.13) | (0.09) | (0.02) | (0.15) |
| Run | 0.94*** | 0.69*** | 0.24*** | 0.90*** |
|  | (0.06) | (0.04) | (0.01) | (0.07) |
| Social:LowOutcome | 0.12 | 2.17*** | 0.69*** | 2.05*** |
|  | (0.18) | (0.13) | (0.03) | (0.22) |
| Social:Run | –0.29*** | 0.00 | 0.18*** | –0.23* |
|  | (0.08) | (0.06) | (0.01) | (0.10) |
| LowOutcome:Run | –0.54*** | 0.41*** | –0.17*** | 0.53*** |
|  | (0.08) | (0.06) | (0.01) | (0.10) |
| Social:LowOutcome:Run | 0.26* | –0.88*** | –0.33*** | –1.21*** |
|  | (0.12) | (0.08) | (0.02) | (0.14) |
| $R^2$ (ord) | 0.255 | 0.220 | 0.063 | 0.218 |
| $R^2$ (adj) | 0.255 | 0.220 | 0.063 | 0.218 |
| N | 20,480 | 61,760 | 947,840 | 17,920 |

**Appendix 1—table 9.** Linear mixed model regressions on BOLD response parameter estimates obtained during the outcome phase – response during *LowOutcomes*.

To identify regions likely to be involved in the guilt effect, we focused on the regions engaged when participants rather than their partner made the choice, that is, the regions responding significantly more to the *Social* than the *Partner* condition. Of these regions, the insulae and the right middle temporal cortex also showed a significant *Social:LowOutcome* interaction. To better understand this interaction in these three ROIs, we ran additional models to test the effect of the *Social* compared to the *Partner* condition on the responses to *Low lottery outcomes* only, and on their response difference between *Low* and *High lottery outcomes*. The results for the response to low lottery outcomes were:

|  | InsulaL | InsulaR | MidTempR |
|---|---|---|---|
| (Intercept) | 0.70*** | 0.95*** | 1.22*** |
|  | (0.18) | (0.18) | (0.18) |
| Social | 0.41*** | 0.18*** | –0.12** |
|  | (0.01) | (0.01) | (0.04) |
| $R^2$ (ord) | 0.150 | 0.148 | 0.231 |
| $R^2$ (adj) | 0.150 | 0.148 | 0.231 |
| N | 186,240 | 253,760 | 12,800 |

**Appendix 1—table 10.** Linear mixed model regressions on BOLD response parameter estimates obtained during the outcome phase – difference *LowOutcome – HighOutcome*.

The results of the models fitted to the response difference (response to *Low lottery outcomes* minus response to *High lottery outcomes*) were:

|  | InsulaL | InsulaR | MidTempR |
|---|---|---|---|
| (Intercept) | –0.30 | –0.07 | –0.48* |
|  | (0.19) | (0.16) | (0.23) |
| Social | 0.44*** | 0.19*** | 0.67*** |
|  | (0.02) | (0.01) | (0.04) |
| $R^2$ (ord) | 0.116 | 0.091 | 0.253 |
| $R^2$ (adj) | 0.116 | 0.091 | 0.253 |
| N | 186,240 | 253,760 | 12800 |

**Appendix 1—table 11.** Participant's judgments of their partner.

|  | Study 1 | Study 2 |
|---|---|---|
| How sympathetic did you find them? | 8.56 (1.7) | 9.18 (1.17) |
| How well could you cooperate with them? | 8.35 (1.29) | 8.68 (1.18) |
| How honest did they seem? | 9.05 (1.11) | 9.34 (1.10) |
| How open were they? | 8.43 (1.89) | 9.05 (1.12) |
| How sociable were they? | 8.63 (1.61) | 9.28 (1.18) |

Scale used was 1 (minimum) to 10 (maximum). Mean and standard deviations are reported.

