## [Editor Report · eLife Assessment]

This manuscript provides **valuable** novel insights into the role of interpersonal guilt in social decision-making by showing that responsibility for a partner's bad lottery outcomes influences happiness. Through the integration of neuroimaging and computational modelling methods, and by combining findings from two studies, the authors provide **solid** support for their claims. The findings will be of interest to researchers in the field of social neuroscience and decision making.

---

## [Referee Report · Reviewer #1 (Public review)]

Summary:

The authors aimed to characterize neurocomputational signals underlying interpersonal guilt and responsibility. Across two studies, one behavioral and one fMRI, participants made risky economic decisions for themselves or for themselves and a partner; they also experienced a condition in which the partners made decisions for themselves and the participant. The authors also assessed momentary happiness intermittently between choices in the task. Briefly, the results demonstrated that participants' self-reported happiness decreased after disadvantageous outcomes for themselves and when both they and their partner were affected; and this effect was exacerbated when participants were responsible for their partner's low outcome, rather than the opposite, reflecting experienced guilt. Consistent with previous work, BOLD signals in the insula correlated with experienced guilt and insula-right IFG connectivity was enhanced when participants made risky choices for themselves and safe choices for themselves and a partner.

Strengths:

This study implements an interesting approach to investigating guilt and responsibility; the paradigm in particular is well-suited to approach this question, offering participants the chance to make risky vs. safe choices that affect both themselves and others. I appreciate the assessment of happiness as a metric for assessing guilt across the different task/outcome conditions, as well as the implementation of both computational models and fMRI.

Weaknesses:

In spite of the overall strengths of the study, I think there are a few areas in which the paper fell a bit short and could be improved.

Comment on the revised submission:

I appreciate the authors' attention to all of my comments and questions regarding the initial version of the paper. However, I still do not believe that the point about the small volume correction in the insula has been adequately addressed. The authors claim that because the SVC was done using an anatomically defined ROI, that it is valid and not double dipping. I understand where the authors are coming from. However, there are a few issues here. First, any use of ROIs is best done via pre-registration (Gentili et al., 2021, European Journal of Neuroscience). Second, the whole set of analyses in this section leading up to the SVC seems somewhat circular. The first step was a whole brain contrast of lottery vs. safe outcomes, which revealed activation in many areas including the insula. Then, it appears that the parameter estimates from the insula were extracted and submitted offline to linear mixed models probing for effects of outcome magnitude, social condition and time, which revealed that the insula activation demonstrated the 'sought after' effect. Next, the manuscript states that the authors attempted to confirm these results with a univariate analysis for the so-called guilt effect within regions showing a stronger response to outcomes of risky relative to safe outcomes, which again showed activation in the insula (not surprisingly), and then a small volume correction was applied to these insula voxels. While an anatomical ROI from a different study was used for the correction, the issue is that multiple analyses already revealed that the insula was involved in the effect of interest. It is unclear why this is even necessary given that the LMM analysis demonstrated the expected result.

---

## [Referee Report · Reviewer #2 (Public review)]

Summary

This manuscript focuses on the role of social responsibility and guilt in social decision making by integrating neuroimaging and computational modeling methods. Across two studies, participants completed a lottery task in which they made decisions for themselves or for a social partner. By measuring momentary happiness throughout the task, the authors show that being responsible for a partner's bad lottery outcome leads to decreased happiness compared to trials in which the participant was not responsible for their partner's bad outcome. At the neural level, this guilt effect was reflected in increased neural activity in the anterior insula, and altered functional connectivity between the insula and the inferior frontal gyrus. Using computational modeling, the authors show that trial by trial fluctuations in happiness were successfully captured by a model including participant and partner rewards and prediction errors (a 'responsibility' model), and model-based neuroimaging analyses suggested that prediction errors for the partner were tracked by the superior temporal sulcus. Taken together, these findings suggest that responsibility and interpersonal guilt influence social decision making.

Strengths

This manuscript investigates the concept of guilt in social decision making through both statistical and computational modeling. It integrates behavioral and neural data, providing a more comprehensive understanding of the psychological mechanisms. For the behavioral results, data from two different studies is included, and although minor differences are found between the two studies, the main findings remain consistent. The authors share all their code and materials, leading to transparency and reproducibility of their methods.

The manuscript is well-grounded in prior work. The task design is inspired by a large body of previous work on social decision making, and includes the necessary conditions to support their claims (i.e., Solo, Social, and Partner conditions). The computational models used in this study are inspired by previous work, and build on well-established economic theories of decision making. The research question and hypotheses clearly extend previous findings, and the more traditional univariate results align with prior work.

The authors conducted extensive analyses, as supported by the inclusion of different linear models and computational models described in the supplemental materials. Psychological concepts like risk preferences are defined and tested in different ways, and different types of analyses (e.g., univariate and multivariate neuroimaging analyses) are used to try to answer the research questions. The inclusion and comparison of different computational models provides compelling support for the claim that partner prediction errors indeed influence task behavior, as illustrated by the multiple model comparison metrics and the good model recovery.

The authors did a good job acknowledging other factors that could differ between the conditions, including the role of other emotions (like empathy) or agency in the decision making process. These additional analyses and nuances strengthen the manuscript and the interpretability of the findings.

Weaknesses

As the authors already note, they did not directly ask participants to report their feelings of guilt. The authors clearly describe this limitation, and also note that in addition to guilt, other emotions like empathy could also be at play in interpersonal decisions. Despite this limitation, this study provides insights into the neural and behavioral mechanisms of responsibility and guilt in social decision making, and how they influence behavior.

---

## [Author Response]

The following is the authors’ response to the original reviews.

**Public Reviews:**

**Reviewer #1 (Public review):**
SummaryThe authors aimed to characterize neurocomputational signals underlying interpersonal guilt and responsibility. Across two studies, one behavioral and one fMRI, participants made risky economic decisions for themselves or for themselves and a partner; they also experienced a condition in which the partners made decisions for themselves and the participant. The authors also assessed momentary happiness intermittently between choices in the task. Briefly, results demonstrated that participants' self-reported happiness decreased after disadvantageous outcomes for themselves and when both they and their partner were affected; this effect was exacerbated when participants were responsible for their partner's low outcome, rather than the opposite, reflecting experienced guilt. Consistent with previous work, BOLD signals in the insula correlated with experienced guilt, and insula-right IFG connectivity was enhanced when participants made risky choices for themselves and safe choices for themselves and a partner.Strengths:This study implements an interesting approach to investigating guilt and responsibility; the paradigm in particular is well-suited to approach this question, offering participants the chance to make risky v. safe choices that affect both themselves and others. I appreciate the assessment of happiness as a metric for assessing guilt across the different task/outcome conditions, as well as the implementation of both computational models and fMRI.

We thank Reviewer 1 for their positive assessment of our manuscript.

Weaknesses:In spite of the overall strengths of the study, I think there are a few areas in which the paper fell a bit short and could be improved.

We thank Reviewer 1 for their comments, which we have used to improve our manuscript. We hope that these changes address the issues raised by the Reviewer.

(1) While the framing and goal of this study was to investigate guilt and felt responsibility, the task implemented - a risky choice task with social conditions - has been conducted in similar ways in past research that were not addressed here. The novelty of this study would appear to be the additional happiness assessments, but it would be helpful to consider the changes noted in risk-taking behavior in the context of additional studies that have investigated changes in risky economic choice in social contexts (e.g., Arioli et al., 2023 Cerebral Cortex; Fareri et al., 2022 Scientific Reports).

We certainly agree that several previously published studies have relied on risky choice tasks with social conditions. In this revised version, we now mention these two studies in the substantially revised Introduction.

(2) The authors note they assessed changes in risk preferences between social and solo conditions in two ways - by calculating a 'risk premium' and then by estimating rho from an expected utility model. I am curious why the authors took both approaches (this did not seem clearly justified, though I apologize if I missed it). Relatedly, in the expected utility approach, the authors report that since 'the number of these types of trials varied across participants', they 'only obtained reliable estimates for [gain and loss] trials in some participants' - in study 1, 22 participants had unreliable estimates and in study 2, 28 participants had unreliable estimates. Because of this, and because the task itself only had 20 gains, 20 losses, and 20 mixed gambles per condition, I wonder if the authors can comment on how interpretable these findings are in the Discussion. Other work investigating loss aversion has implemented larger numbers of trials to mitigate the potential for unreliable estimates (e.g., Sokol-Hessner et al., 2009).

We agree that we have not clearly justified why we have taken two approaches to assess risk preferences. In short, while the expected utility approach is a more comprehensive method to model a participant’s choices, we had not sufficiently considered the need for the large number of trials required to fit such models when designing our experiment. Calculating the risk premium was the less comprehensive, simpler alternative that we could calculate for all participants. We have now mentioned this fact in the Results section. As the only difference in risk aversion across conditions was found in Study 1 using the expected utility method, which could only be successfully applied in a minority of participants, we believe that this difference should not be taken as a strong finding. We have now mentioned this fact in the revised Discussion.

(3) One thing seemingly not addressed in the Discussion is the fact that the behavioral effect did not replicate significantly in study 2.

We agree that we had not sufficiently discussed the fact that there were (slight but significant) differences in risk preferences between the Solo and Social conditions in Study 1 but not in Study 2. We now do so in the revised Discussion, and write the following:

“Participants made slightly more risk-seeking choices when deciding for themselves than for both themselves and the partner in Study 1, but this difference disappeared in Study 2. The *ρ* parameter on which this finding in Study 1 is based could only be estimated in a minority of participants due to a relatively low number of trials, which suggests that this finding may not be very reliable. The simpler and more robust method (evaluation of a risk premium) showed no difference in risk aversion across conditions in either study. Overall, we believe that we do not have strong evidence of differences in risk preferences across conditions.”

(4) Regarding the computational models, the authors suggest that the Reponsibility and Responsibility Redux models provided the best fit, but they are claiming this based on separate metrics (e.g., in study 1, the redux model had the lowest AIC, but the responsibility only model had the highest R^2; additionally, the basic model had the lowest BIC). I am wondering if the authors considered conducting a direct model comparison to statistically compare model fits.

We agree that we should run formal, direct model comparison tests. We now ran likelihood-ratio tests which showed that the Responsibility model was the best. We now report this in the Results section, just below Table 1:

“A likelihood ratio test (Equation 9) revealed that the Responsibility model fitted better than all the other models, including the *Responsibility Redux* model (Study 1: all *LR* ≥ 47.36, *p* < 0.0001; Study 2: all *LR* ≥ 77.83, *p* < 0.0001).”

(5) In the reporting of imaging results, the authors report in a univariate analysis that a small cluster in the left anterior insula showed a stronger response to low outcomes for the partner as a result of participant choice rather than from partner choice. It then seems as though the authors performed small volume correction on this cluster to see whether it survived. If that is accurate, then I would suggest that this result be removed because it is not recommended to perform SVC where the volume is defined based on a result from the same whole-brain analysis (i.e., it should be done a priori).

As indicated in the manuscript, the small insula cluster centered at [-28 24 -4] and shown in Figure 4F survived corrections for multiple tests within the anatomically-defined anterior insula (based on the anatomical maximum probability map described in Faillenot et al., 2017), which is independent of the result of our analysis. Functionally defining the small volume based on the same data would indeed be circular and misleading “double-dipping”. We have most certainly NOT done this. The reason why we selected the anterior insula is because it is one of the regions most frequently associated with guilt (see the explanations in our Introduction, which refers for example to Bastin et al., 2016; Lamm & Singer, 2010; Piretti et al., 2023). Thus we feel that performing small-volume correction within the anatomically-defined anterior insula is a valid analysis. We fully acknowledge that, independently of any correction, the effect and the cluster are small. We now write:

“We found a weak response in a small cluster within the left anterior insula (peak *T* = 3.95, d = 0.59, 22 voxels, peak intensity at [-28 24 -4]; Figure 4F). Given the documented association between anterior insula and guilt (see Introduction), we proceeded to test whether this result survived correction for family-wise errors due to multiple comparisons restricted to the left anterior insula gray matter [defined anatomically and thus independently from our findings, as the anterior short gyrus, middle short gyrus, and anterior inferior cortex in an anatomical maximum probability map (Faillenot et al., 2017)]. This correction resulted in a p value of 0.024. This result, although it is only a small effect in a small cluster, is consistent with the mixed model analysis reported earlier.”

**Reviewer #2 (Public review):**
SummaryThis manuscript focuses on the role of social responsibility and guilt in social decision-making by integrating neuroimaging and computational modeling methods. Across two studies, participants completed a lottery task in which they made decisions for themselves or for a social partner. By measuring momentary happiness throughout the task, the authors show that being responsible for a partner's bad lottery outcome leads to decreased happiness compared to trials in which the participant was not responsible for their partner's bad outcome. At the neural level, this guilt effect was reflected in increased neural activity in the anterior insula, and altered functional connectivity between the insula and the inferior frontal gyrus. Using computational modeling, the authors show that trial-by-trial fluctuations in happiness were successfully captured by a model including participant and partner rewards and prediction errors (a 'responsibility' model), and model-based neuroimaging analyses suggested that prediction errors for the partner were tracked by the superior temporal sulcus. Taken together, these findings suggest that responsibility and interpersonal guilt influence social decision-making.StrengthsThis manuscript investigates the concept of guilt in social decision-making through both statistical and computational modeling. It integrates behavioral and neural data, providing a more comprehensive understanding of the psychological mechanisms. For the behavioral results, data from two different studies is included, and although minor differences are found between the two studies, the main findings remain consistent. The authors share all their code and materials, leading to transparency and reproducibility of their methods.The manuscript is well-grounded in prior work. The task design is inspired by a large body of previous work on social decision-making and includes the necessary conditions to support their claims (i.e., Solo, Social, and Partner conditions). The computational models used in this study are inspired by previous work and build on well-established economic theories of decision-making. The research question and hypotheses clearly extend previous findings, and the more traditional univariate results align with prior work.The authors conducted extensive analyses, as supported by the inclusion of different linear models and computational models described in the supplemental materials. Psychological concepts like risk preferences are defined and tested in different ways, and different types of analyses (e.g., univariate and multivariate neuroimaging analyses) are used to try to answer the research questions. The inclusion and comparison of different computational models provide compelling support for the claim that partner prediction errors indeed influence task behavior, as illustrated by the multiple model comparison metrics and the good model recovery.

We thank the reviewer very much for their comprehensive description of our study and the positive assessment of our study and approach.

WeaknessesAs the authors already note, they did not directly ask participants to report their feelings of guilt. The decrease in happiness reported after a bad choice for a partner might thus be something else than guilt, for example, empathy or feelings of failure (not necessarily related to guilt towards the other person). Although the patterns of neural activity evoked during the task match with previously found patterns of guilt, there is no direct measure of guilt included in the task. This warrants caution in the interpretation of these findings as guilt per see.

We fully agree that not directly asking participants about feelings of guilt is a clear limitation of our study. While we already mention this in our Discussion, we have expanded our discussion of the consequences on the interpretation of our results along the lines described by the reviewer in the revised manuscript. We would like to thank the reviewer for proposing these lines of thought, and have now made the following changes to the text:

In the first paragraph of the discussion, we now write: “Being responsible for choosing a lottery that yielded a low outcome for a partner made our participants feel worse than witnessing the same outcome resulting from their partner’s choice, which we interpret as interpersonal guilt; although we note that we have not asked participants specifically about which emotion they felt in these situations.

Later on, in the third paragraph focusing on the anterior insula, we now write: “This replicates a large body of evidence associating aIns with feelings of guilt evoked during social decisions (see Introduction). Because we have neither asked our participants specifically what they felt in these situations, nor specifically whether they experienced guilt, we cannot exclude the possibility that they have instead or in addition felt empathy for their partner, a feeling of failure or bad luck, or some other emotion.”

As most comparisons contrast the social condition (making the decision for your partner) against either the partner condition (watching your partner make their decision) or the solo condition (making your own decision), an open question remains of how agency influences momentary happiness, independent of potential guilt. Other open questions relate to individual differences in interpersonal guilt, and how those might influence behavior.

How agency influences momentary happiness or variations thereof during the course of an experiment such as ours is an interesting question in itself. We now ran linear mixed models assessing agency (i.e. we compared happiness in conditions Solo & Social conditions vs. Partner condition), which revealed lower happiness in Solo and Social conditions (i.e. when it was the participant’s turn to decide) in both studies. This is interesting in itself and may reflect the drive behind responsibility aversion reported by Edelson et al.’s 2018 study: being assigned the role of the decider in a social setting may make people slightly unhappy, perhaps due to “weight of the responsibility”. We now report these findings in the Results section, including this proposed explanation; because we were not specifically interested in responsibility aversion, we do not discuss this further in the Discussion. The edited text is under the new subsection entitled ‘Momentary happiness: effects of agency, responsibility and guilt’, on page 12:

“Next, we assessed whether happiness varied depending on the participant’s agency (Social + Solo vs. Partner), and found happiness to be lower when the participant chose, independent of the outcome (Study 1: *t*(3600) = -3.92, *p* = 0.00009, β = -0.14, 95% CI = [-0.20 -0.07]; Study 2: *t*(2870) = -6.07, *p* = 0.000000001, β = -0.24, 95% CI = [-0.31 -0.16]). . This is interesting in itself and may reflect the drive behind responsibility aversion reported by Edelson et al.’s 2018 study: being assigned the role of the decider in a social setting may make people slightly unhappy, perhaps due to “weight of the responsibility”. To specifically search for a sign of interpersonal guilt, [...]”

Regarding individual differences: this is a very interesting topic that we have not addressed here due to the (relatively) small number of participants in our studies, but we might consider this for future follow-up studies, which we mention in the Discussion paragraph regarding open questions.

This manuscript is an impressive combination of multiple approaches, but how these different approaches relate to each other and how they can aid in answering slightly different questions is not very clearly described. The authors could improve this by more clearly describing the different methods and their added value in the introduction, and/or by including a paragraph on implications, open questions, and future work in the discussion.

We thank the reviewer for their appreciation of our complementary approach, and agree that we had not sufficiently explained the reasons why we used several methods. We have now added a paragraph explaining this at the end of the Introduction (page 5):

“We analysed our behavioural data using several complementary methods: choices were modelled with mixed-effects regressions serving as manipulation checks; risk preferences expressed in choices were assessed using a comprehensive expected utility model as well as with a simpler, more robust “risk premium” approach; and happiness data were fitted, in addition to the computational models, with several linear mixed models to assess the impact of both the participant’s and their partner’s rewards, the impact of agency and their interactions. Inspired by findings reported in previous neuroimaging of social emotions, we also used several methods to analyse our fMRI data, including conventional methods (both region-of-interest and mass univariate); mixed-effects regression models; computational model-based analyses (inspired by e.g. Konovalov et al., 2021; Rutledge et al., 2014); and functional connectivity (e.g. Edelson et al., 2018; Konovalov et al., 2021). The behavioural modelling is thus complemented by neuroimaging analyses that offer insight about both the activity in regions associated with guilt as well as their place in a wider network, providing an in-depth comprehensive analysis of the mechanisms behind guilt evoked by social responsibility.”

In addition, as suggested we added the following paragraph on open questions and future work in the Discussion:

“Several open questions remain at the end of this study. As discussed above, asking participants directly about which emotions they have felt during the different stages of this task would allow us to link subjective experience with our analytical measures. Testing more participants would allow us to assess the impact of inter-individual variations in personality traits on the experience as well as the behavioural and neural correlates of guilt and responsibility. Using more trials in the experiment would allow separate modelling of risk preferences in gain and loss trials in each experimental condition using expected utility models, and could allow testing whether changes in momentary happiness affect subsequent choices. Varying partner identities (friends, strangers, artificial agent) could reveal the impact of social discounting on guilt and responsibility. In sum, we believe that this experimental approach lends itself very well to the study of several aspects of social emotions.”

However, taken together, this study provides useful insights into the neural and behavioral mechanisms of responsibility and guilt in social decision-making and how they influence behavior.

We thank the reviewer again for their appreciation of our work and hope that our revisions improved the manuscript.

**Recommendations for the authors:**

**Reviewer #1 (Recommendations for the authors):**
The majority of my suggestions are in the public review, so I will not repeat them here. But in general, I like the paper, and in addition to my other comments, I think that there should be more discussion of the potential limitations of the study and conclusions that can be drawn. I also thought parts of the results were a little hard to follow, particularly in the 'momentary happiness' section. Perhaps an additional subsection here might help with flow.

We agree that we could have discussed further the limitations of our study and the conclusions that can be drawn from it, which we have now done in the last paragraphs of the Discussion in this revised version.

To improve the structure of the section on ‘momentary happiness’, we separated this section into two, entitled: ‘Momentary happiness: links to reward‘ and ‘Momentary happiness: effects of agency, responsibility and guilt’, which should facilitate the reading of this long section. We proceeded in a similar manner for the Choices section, which is now subdivided into ‘Choices: manipulation check’ and ‘Choices: risk preferences’. We believe that these changes have indeed improved the readability of our manuscript.

**Reviewer #2 (Recommendations for the authors):**
Overall, I believe this manuscript was well-designed, consists of extensive analyses, and provides interesting new insights into the mechanisms underlying social decision-making. I mostly have some clarifying questions and minor comments, which are described below.(1) Integration of prior findings in the first paragraphs of the Introduction. Although all the previous work described in the 2nd-5th paragraph introduction is interesting, it felt a bit like an enumeration of findings rather than an integrated introduction leading to the current research question. At the end of paragraph 5, it becomes clear how these findings relate to the current research question, but I believe it will improve the flow and readability of the introduction if this becomes clear earlier on.

We agree that we could have integrated the cited previous work into the Introduction so that the text builds up to the research question. We have now extensively reworked several paragraphs in the Introduction (pages 3-5) and hope that these changes have made it easier to follow.

(2) For the risk attitudes (Choices), you describe pooling the gains and losses and then comparing the social and solo conditions. I was wondering whether you also looked at potential differences between gains and losses (delta measure) for social versus the solo condition (so a comparison of the delta). Based on prior work, I can imagine that the difference in risk attitudes for gains and losses might differ when making decisions for yourself versus when you're doing it for a partner. In general, I was wondering how you explain these findings, as there is also a lot of work showing differences in risk-taking patterns for gains and losses.

We agree that we could have compared delta measures between solo and social conditions. However, as we describe in the Results section and comment on in the Discussion, the relatively low number of trials made separate fitting of gain and loss trials across conditions difficult. While this question could thus be addressed in subsequent versions of our experiment with more trials, such a fine-grained analysis of the decisions was not the focus of our current study.

(3) On page 11, you state: "in particular the partner's reward prediction errors resulting from the participants' decisions, i.e. those pRPE for which participants were responsible." From the results described in the paragraph above, this doesn't become clear (e.g., there's no distinction made between social_pRPE and partner_pRPE in the text), as it only discusses differences in weights between pRPE and sRPE. I would recommend including some more information in the main text on these main modeling findings, so one doesn't have to go to the Supplemental Materials to understand them.

We did indeed fail to report these findings in the text! We thank the reviewer for pointing this out. We have now edited this passage as follows:

“Crucially, we find here that the partner’s reward prediction errors (social_pRPE and partner_pRPE) contributed to explaining changes in participants’ momentary happiness: the Responsibility and ResponsibilityRedux models explained the data better than the models without these parameters (see Table 1). In particular, the partner’s reward prediction errors resulting from the participants’ decisions (social_pRPE), i.e. those pRPE for which participants were responsible, contributed to explaining our data (weights for social_pRPE were greater than 0: Responsibility model: Study 1: *Z* = 2.85, *p* = 0.004, Study 2: *Z* = 3.26, *p* = 0.001; *Responsibility Redux* model: Study 1: *Z* = 2.93, *p* = 0.003, Study 2: *Z* = 3.30, *p* = 0.001; weights for social_pRPE tended to be higher than weights for partner_pRPE: Responsibility model: Study 1: *Z* = 2.14, *p* = 0.033; Study 2: *Z* = 1.41, *p* = 0.16).”

(4) The functional connectivity findings seem to come out of nowhere and are not introduced or described anywhere prior in the manuscript. It is therefore not completely clear why you conducted these analyses, or what they add above and beyond previous analyses. Already introducing this method earlier on would fix that.

We agree that we could have introduced functional connectivity analyses earlier in the text, particularly given the many previous studies in our field using this technique. We have now done this at the end of a new last paragraph of the Introduction:

“Inspired by findings reported in previous neuroimaging of social emotions, we also used several methods to analyse our fMRI data, including conventional methods (both region-of-interest and mass univariate); mixed-effects regression models; computational model-based analyses (inspired by e.g. Konovalov et al., 2021; Rutledge et al., 2014); and functional connectivity (e.g. Edelson et al., 2018; Konovalov et al., 2021). The behavioural modelling is thus complemented by neuroimaging analyses that offer insight about both the activity in regions associated with guilt as well as their place in a wider network, providing an in-depth comprehensive analysis of the mechanisms behind guilt evoked by social responsibility.”

(5) For the functional connectivity findings: I was wondering why you only looked at the choice phase, and not at the feedback phase. I understand that previous work focused on the choice phase, but for the purpose of this study (focus on guilt), I can imagine it is also interesting to see what happens with feedback. In the discussion, you also state "How we feel when we witness our decisions' consequences on others is an important signal to consider when attempting to make good social decisions." (p. 19), which is more focused on the feedback rather than choice, and also supports the idea that looking at the feedback moment might be relevant.

We agree that we could also have looked at the functional connectivity during the feedback phase. The main reason why we had originally not done so was time constraints. At the current time we would in addition point out that the manuscript is already very long and contains many analyses of behavioural and fMRI data. Adding this analysis would cost additional time and would further delay the publication of our manuscript, which we would prefer to avoid. However, one could of course look at these effects in subsequent analyses of the same data or in subsequent versions of this experiment. We have now mentioned this in the Discussion, in the paragraphs on open questions.

Minor comments:(1) For some of the Figures, it would be helpful if the subtitles were more informative. For Figure 2 and Figure 3 for example, it would be nice if Study 1 and Study 2 were not only mentioned in the figure description but also in the actual figure. For Figures 3 and 4, it would be helpful to have significance stars for the bar plots as well.

We agree that these changes make the figures more easily understandable and have implemented them all, except for adding stars on Figure 4, because all bar plots in panels C and E would have been labeled with two or more stars, which would have made the figure difficult to read. We have now mentioned the fact that all these coefficients were significant in the figure legend.

(2) For some of the Supplementary Results, it would be very helpful if there was a legend or description. This is already the case for most of the SR, but not for all.

We have now added a legend to all elements of the Supplementary Results.

Some questions that came to mind while going through them:- Supplementary Table 1: which p-values correspond to the significance stars? This information is included for Supplementary Table 2, but not for ST1.

We have now added the missing information in ST1.

- Supplementary Figure 1: do the colors correspond to different participants?

We have now specified that the colors do indeed correspond to different participants.

- Supplementary Table 5 (final table): what do the - represent? As in, why is there no value for "run" for the MPFC? At first, I thought you only included the significant values, but then I noticed a few non-significant values as well, so it wasn't completely clear to me why some of the values were missing. This also applies to Supplementary Table 6.

We have indeed forgotten to explain this. The ‘-’ in Supplementary Tables 4 and 6 indicate that the linear mixed model without the factor ‘run’ was the better-fitting one. We have now added the following explanation in the text accompanying Supplementary Table 4:

“We tested these models both with and without the factor Run and associated interaction, and we report the best-fitting model in the table below: a dash (‘-’) in the row displaying parameters for the run and socialVsSolo:run regressors indicates that the model without factor run was better-fitting for this ROI.”

(3) I came across a few minor typos or sentences that were not completely clear to me.- On page 3: "Patients with damage to ventromedial prefrontal cortex (vmPFC) seem insensitive to guilt when playing social economic games (Krajbich et al., 2009)." This sentence felt a bit out of nowhere and doesn't logically follow from the previous sentences.

We have now revised the descriptions of this previous study as well as several others and how they fit into the research question.

- On page 3: "In another study, participant errors in a difficult perception task lead to a partner feeling pain and evoked activations in left aIns and dlPFC (Koban et al., 2013)." This sentence doesn't really flow, and from the wording, it is not completely clear whether it's the errors or the partner pain that led to the aIns and dlPFC activation.

We have now revised the description of this study as well, as follows:

“In another study, partners received painful stimuli when participants made errors during a difficult perception task. These errors evoked activations in the left aIns and dlPFC in the participants (Koban et al., 2013).”

- Supplementary Figure 1: there is a missing period after the sentence "We then compared these new estimated parameters to the actual parameters from which the synthetic data were generated"

We have now added a missing comma after “generated”.

- On page 5: "We ran two experiments, Study 1 outside fMRI and Study 2 during fMRI, with separate groups of participants." I would change "outside fMRI" to outside the MRI scanner or something like that, as it's not completely correct to say "outside fMRI".

We have changed the sentence to “outside the MRI scanner”.

- On page 6: for the first result, there are currently two p-values reported (p < 2.5e-20 and p < 2e-16). I believe this is an error?

This was indeed an error! We have re-run this analysis, noticed that also the degrees of freedom were miscalculated, and have updated this result and the effect of condition (solo vs social). Results are almost identical as previously and all conclusions hold. We have also checked the other analyses reported in this paragraph – all results replicate exactly.

- On page 6: "Supplemental Table 1" should be "Supplementary Table 1" (for consistency).

Done.

On page 8: "participants in both conditions of both studies", I would change "of both studies" to "for both studies".

Done.

On page 8: for the "Momentary Happiness" paragraph, it would be helpful if you could briefly describe the Rutledge method here, for people who are unfamiliar with the approach.

We now write the following at the beginning of this paragraph:

“Following Rutledge and colleagues’ methodology, which considers that changes in momentary happiness in response to outcomes of a probabilistic reward task are explained by the combined influence of recent reward expectations and prediction errors arising from those expectations, we fitted computational models to each participant’s happiness data.”

On page 10: "Wilkoxon sign-rank tests", should be "Wilcoxon".

Done.

We thank the reviewer for their careful reading of our manuscript. We believe that these changes have indeed improved our manuscript.